# EnsIR: An Ensemble Algorithm for Image Restoration via Gaussian Mixture Models

**Shangquan Sun**[1,2]     **Wenqi Ren**[3,4*]     **Zikun Liu**[5]
**Hyunhee Park**[6]     **Rui Wang**[1,2]     **Xiaochun Cao**[3]

[1]Institute of Information Engineering, Chinese Academy of Sciences, Beijing, China
[2]School of Cyber Security, University of Chinese Academy of Sciences, Beijing, China
[3]School of Cyber Science and Technology, Shenzhen Campus of Sun Yat-sen University
[4]Guangdong Provincial Key Laboratory of Information Security Technology
[5]Samsung Research China - Beijing (SRC-B)
[6]Camera Innovation Group, Samsung Electronics
{sunshangquan,wangrui}@iie.ac.cn
{zikun.liu,inextg.park}@samsung.com
{renwq3,caoxiaochun}@mail.sysu.edu.cn

## Abstract

Image restoration has experienced significant advancements due to the development of deep learning. Nevertheless, it encounters challenges related to ill-posed problems, resulting in deviations between single model predictions and ground-truths. Ensemble learning, as a powerful machine learning technique, aims to address these deviations by combining the predictions of multiple base models. Most existing works adopt ensemble learning during the design of restoration models, while only limited research focuses on the inference-stage ensemble of pre-trained restoration models. Regression-based methods fail to enable efficient inference, leading researchers in academia and industry to prefer averaging as their choice for post-training ensemble. To address this, we reformulate the ensemble problem of image restoration into Gaussian mixture models (GMMs) and employ an expectation maximization (EM)-based algorithm to estimate ensemble weights for aggregating prediction candidates. We estimate the range-wise ensemble weights on a reference set and store them in a lookup table (LUT) for efficient ensemble inference on the test set. Our algorithm is model-agnostic and training-free, allowing seamless integration and enhancement of various pre-trained image restoration models. It consistently outperforms regression-based methods and averaging ensemble approaches on 14 benchmarks across 3 image restoration tasks, including super-resolution, deblurring and deraining. The codes and all estimated weights have been released in Github.

## 1   Introduction

Image restoration has witnessed significant progress over the decades, especially with the advent of deep learning. Numerous architectures have been developed to solve the problem of restoration, including convolutional neural networks (CNNs) [10, 82], vision Transformers (ViTs) [39, 81, 93] and recently, vision Mambas [27]. However, single models with different architectures or random initialization states exhibit prediction deviations from ground-truths, resulting in sub-optimal restoration results.

---

*Corresponding Author

38th Conference on Neural Information Processing Systems (NeurIPS 2024).

To alleviate this problem, ensemble learning, a traditional but influential machine learning technique, has been applied to image restoration. It involves combining several base models to obtain a better result in terms of generalization and robustness [18, 20, 26, 49, 57]. However, most ensemble methods in image restoration focus on training-stage ensemble requiring the ensemble strategy to be determined while training multiple models, thus sacrificing flexibility of changing models and convenience for plug-and-play usage [31, 35, 40, 44, 46, 54, 59, 70, 76]. In contrast, there is a demand for advanced post-training ensemble methods in the image restoration industry, where researchers still prefer averaging as their primary choice [1, 15, 41, 52, 66, 68, 92].

Despite the industrial demand, post-training ensemble in image restoration is challenging for tradition ensemble algorithms originally designed for classification or regression. Unlike classification and regression, image restoration predictions are matrices with each pixel is correlated with others and range from 0 to 255. As a result, traditional methods like bagging [6] and boosting [28] either require enormous computational resources for the restoration task or fail to generalize well due to the imbalance between candidate number and feature dimension. As an alternative, Jiang *et al.* propose a post-training ensemble algorithm for super-resolution by optimizing a maximum a posteriori problem with a reconstruction constraint [31]. However, this constraint requires an explicit expression of the degradation process, which is extremely difficult to define for other restoration tasks beyond super-resolution. It also necessitates prior knowledge of the base models' performance, further limiting its practical application. These issues with both traditional and recent ensemble methods lead researchers in image restoration to prefer weighted averaging as their primary ensemble approach [1, 15, 52, 68, 92].

To this end, we formulate the ensemble of restoration models using Gaussian mixture models (GMMs), where ensemble weights can be efficiently learned via the expectation maximization (EM) algorithm and stored in a lookup table (LUT) for subsequent inference. Specifically, we first base on the Gaussian prior that assumes the prediction error of each model follows a multivariate Gaussian distribution. Based on the prior, the predictions of multiple samples can be appended into a single variable following Gaussian. By partitioning pixels into various histogram-like bins based on their values, we can convert the problem of ensemble weight estimation into various solvable univariate GMMs. As a result, the univariate GMMs can be solved to obtain range-wise ensemble weights by the EM algorithm, with the means and variances of Gaussian components estimated based on the observation priors. We estimate these weights on a reference set and store them in a LUT for efficient inference on the test set. Our method does not require training or prior knowledge of the base models and degradation processes, making it applicable to various image restoration tasks.

Our contributions mainly lie in three areas:

- Based on the Gaussian prior, we partition the pixels of model predictions into range-wise bin sets of mutually exclusive ranges and derive the ensemble of multi-model predictions into weight estimation of various solvable univariate GMMs.

- To solve the univariate GMMs and estimate ensemble weights, we leverage the EM algorithm with means and variances initialized by observed prior knowledge. We construct a LUT to store the range-wise ensemble weights for efficient ensemble inference.

- Our ensemble algorithm does not require extra training or knowledge of the base models. It outperforms existing post-training ensemble methods on 14 benchmarks across 3 image restoration tasks, including super-resolution, deblurring and deraining.

## 2 Related Works

**Ensemble Methods.** Ensemble methods refer to approaches that fuse the predictions of multiple base models to achieve better results than any of individual model [20]. Traditional ensemble methods include bagging [6], boosting [28], random forests [7], gradient boosting [24], histogram gradient boosting [14, 34], etc. These methods have been applied to various fields in classification and regression, such as biomedical technology [77, 78], intelligent transportation [53, 84], and pattern recognition [61, 91].

**Image restoration.** Image restoration, as a thriving area of computer vision, has been making significant progress since the advent of deep learning [10, 65]. Various model architectures have been

proposed to address image restoration tasks, such as convolutional neural networks (CNNs) [19, 56, 73, 90], multilayer perceptron (MLPs) [67], vision Transformers (ViTs) [39, 62, 81], etc. Additionally, various structural designs like multi-scale [56], multi-patch [64, 89], and progressive learning [82] have been adopted to improve the representative capacity. It is known that CNNs excel at encoding local features, while ViTs are adept at capturing long range dependencies. Despite this significant progress, single models still generate predictions that deviate from ground-truths, leading researchers in industry to adopt multi-model ensembles to achieve better performance [15, 52, 68, 92].

**Ensemble Learning in Image Restoration.** Some works incorporate ensemble learning into image restoration by training multiple networks simultaneously and involving ensemble strategy during the training process [2, 3, 9, 11–13, 16, 17, 21, 29, 31, 33, 35–38, 40, 42–46, 54, 55, 59, 63, 69, 70, 72, 75, 76, 79, 80, 85, 87]. However, the majority of them require additional training or even training from scratch alongside ensemble. Only a few focus on ensemble at the post-training stage [31, 66]. Among them, self-ensemble [66] geometrically augments an input image, obtains super-resolution predictions of augmented candidates and applies averaging ensemble to the candidates, which is orthogonal to the scope of our work. RefESR [31] requires a reconstruction objective which must get access to the degradation function, prohibiting its application to tasks other than super-resolution. In general, limited works focus on the training-free ensemble of image restoration. There lacks a general ensemble algorithm for restoration despite industry demand [1, 15, 52, 68, 92].

# 3 Proposed Ensemble Method for Image Restoration

In Sec. 3.1, we first present the formulation of the ensemble problem in image restoration. We then formulate our ensemble method in the format of Gaussian mixture models (GMMs) over partitioned range-wise bin sets in Sec. 3.2. We derive the expectation maximization (EM) algorithm with known mean and variance as prior knowledge to solve the GMMs problems in Sec. 3.3

## 3.1 Ensemble Formulation of Image Restoration

Given a test set $\mathbb{T} = \{\hat{\mathbf{X}}_n, \hat{\mathbf{Y}}_n\}$ with numerous pairs of input images and ground-truths, suppose we have $M$ pre-trained base models for image restoration, $f_1, ..., f_M$. For a model $f_m$ where $m \in \{1, ..., M\}$, its prediction is denoted by $\tilde{\mathbf{X}}_{m,n} = f_m(\hat{\mathbf{X}}_n)$ for abbreviation. We consider the ensemble of the predictions as a weighted averaging, i.e.,

$$\tilde{\mathbf{Y}}_n = \boldsymbol{\beta}_n^\top \begin{bmatrix} \tilde{\mathbf{X}}_{1,n} & \cdots & \tilde{\mathbf{X}}_{M,n} \end{bmatrix}, \ \forall n \tag{1}$$

where $\tilde{\mathbf{Y}}_n$ is the ensemble result, and $\boldsymbol{\beta}_n \in \mathbb{R}^M$ is the vector of weight parameters for the ensemble. The widely-used averaging ensemble strategy in image restoration assigns equal weights for all samples and pixels, i.e., $\boldsymbol{\beta}_n = \begin{bmatrix} \frac{1}{M} & \cdots & \frac{1}{M} \end{bmatrix}$. A recent method in the NTIRE 2023 competition assigns weights inversely proportional to the mean squared error between the predictions and their average [92]. However, they adopt globally constant weights for all pixels and samples, neglecting that the performances of base models may fluctuate for different patterns and samples.

Alternatively, we start from the prospective of GMMs and assign range-specific weights based on the EM algorithm.

## 3.2 Restoration Ensemble as Gaussian Mixture Models

Similar to RefESR [31], suppose we have a reference set $\mathbb{D} = \{\mathbf{X}_n, \mathbf{Y}_n\}_{n=1}^N$ with $N$ pairs of input images and ground-truths. We assume the reference set and test set are sampled from the same data distribution, i.e., $\mathbb{D}, \mathbb{T} \sim \mathcal{D}$.

For each model $f_m$, its prediction is denoted by $\mathbf{X}_{m,n} = f_m(\mathbf{X}_n) \in \mathbb{R}^{3 \times H \times W}$. We use $\mathbf{x}_{m,n}, \mathbf{y}_n \in \mathbb{R}^L$ to represent the flattened vector of the matrices $\mathbf{X}_{m,n}, \mathbf{Y}_n$, where $L = 3HW$. Based on Gaussian prior, it can be assumed that the estimation error of a model on an image follows a zero-mean Gaussian distribution, namely $\boldsymbol{\epsilon}_{m,n} = \mathbf{y}_n - \mathbf{x}_{m,n} \sim \mathcal{N}(\mathbf{0}, \boldsymbol{\Sigma}_{m,n})$, where $\boldsymbol{\Sigma}_{m,n} \in \mathbb{R}^{L \times L}$ is the covariance matrix of the Gaussian. Then the observed ground-truth can be considered following a multivariate Gaussian with the mean equal to the prediction, i.e., $\mathbf{y}_n | f_m, \mathbf{x}_n \sim \mathcal{N}(\mathbf{x}_{m,n}, \boldsymbol{\Sigma}_{m,n})$.

We can consider the ensemble problem as the weighted averaging of Gaussian variables and estimate the weights by solving its maximum likelihood estimation. However, solving the sample-wise mixture

---

**Algorithm 1:** EnsIR: an ensemble algorithm for image restoration

---

**Input:** A small reference dataset $\{\mathbf{x}_n, \mathbf{y}_n\}_{n=1}^N$ for ensemble weight estimation, test set $\{\hat{\mathbf{X}}_n\}$, $M$ pre-trained models $f_1, ..., f_M$, bin width $b$, Empty lookup table LUT

**Output:** Ensemble result $\{\tilde{\mathbf{Y}}_n\}$

**Estimation Stage:**

1   Obtain restoration predictions by $\mathbf{x}_{m,n} = \text{flatten}\left(f_m(\mathbf{X}_n)\right)$, $\forall m \in \{1, ..., M\}$ ;

2   Append restoration predictions and ground-truths into $\mathbf{y}_{1:N}$ and $\mathbf{x}_{m,1:N}$ based on Eq. 2 ;

3   Define bin set space $\mathbb{B} = \{[0, b), [b, 2b), ..., [(T-1)b, 255]\}$ ;

4   **for** *each bin set* $(B_1, ...B_M) \in \mathbb{B}^M$ **do**

5      Compute the partition map $\mathbf{R}_r = \prod_{m=1}^M \mathbf{I}_{B_m}(f_m(\mathbf{x}_n))$ ;

6      Partition images and obtain range-wise patches $(\mathbf{y}_{r,1:N}, \mathbf{x}_{r,1,1:N}, ..., \mathbf{x}_{r,M,1:N})$ by Eq. 3 ;

7      $(\alpha_{r,1}, ..., \alpha_{r,M}) \leftarrow \textbf{MPEM}(\mathbf{y}_{r,1:N}, \mathbf{x}_{r,1,1:N}, ..., \mathbf{x}_{r,M,1:N})$ ;

8      Store LUT$[(B_1, ...B_M)] \leftarrow (\alpha_{r,1}, ..., \alpha_{r,M})$ ;

9   **end**

**Inference Stage:**

10   **for** *each test data* $\hat{\mathbf{X}}_n$ **do**

11      **for** *each bin set* $(B_1, ...B_M) \in \mathbb{B}^M$ **do**

12          Retrieve $(\alpha_{r,1}, ..., \alpha_{r,M}) \leftarrow \text{LUT}[(B_1, ...B_M)]$ ;

13          Partition input as $\tilde{\mathbf{X}}_{r,m,n} \leftarrow \mathbf{R}_r \cdot f_m(\hat{\mathbf{X}}_n)$, where $\mathbf{R}_r = \prod_{m=1}^M \mathbf{I}_{B_m}(f_m(\hat{\mathbf{X}}_n))$ ;

14          $\tilde{\mathbf{Y}}_{r,n} \leftarrow \sum_{m=1}^M \alpha_{r,m} \tilde{\mathbf{X}}_{r,m,n}$ ;        /* Inner summation of Eq. 15 */

15      **end**

16      $\tilde{\mathbf{Y}}_n \leftarrow \sum_{r=1}^{T^M} \tilde{\mathbf{Y}}_{r,n}$ ;             /* Outer summation of Eq. 15 */

17   **end**

---

of Gaussian is not feasible because the covariance matrices are sample-wise different and thus hard to estimate. Besides, the number of prediction samples is much fewer than feature dimension, resulting in the singularity of the covariance matrices. Please refer to Sec. A.1 in Appendix for details.

In contrast, we alternatively append the reference set into a single sample following Gaussian as

$$\mathbf{y}_{1:N}|f_m, \mathbf{x}_{1:N} \sim \mathcal{N}\left(\mathbf{x}_{m,1:N}, \text{diag}(\boldsymbol{\Sigma}_{m,1}, ..., \boldsymbol{\Sigma}_{m,N})\right), \tag{2}$$

where $\mathbf{y}_{1:N} = [\mathbf{y}_1 \; \cdots \; \mathbf{y}_N] \in \mathbb{R}^{NL}$ and $\mathbf{x}_{m,1:N} = [\mathbf{x}_{m,1} \; \cdots \; \mathbf{x}_{m,N}] \in \mathbb{R}^{NL}$ are the concatenation of observed ground-truths and restored samples respectively. Since data samples can be considered following *i.i.d* data distribution $\mathcal{D}$, the variance of the concatenated samples is diagonal.

However, the covariance matrix is still singular due to the imbalance between prediction sample number and feature dimension. Thus, directly mixing the multivariate Gaussian is still infeasible to solve. We thus alternatively categorize pixels into various small bins of mutually exclusive ranges such that the pixels within each range can be considered following a univariate Gaussian distribution according to the central limit theorem. Concretely, we separate the prediction range of models into $T$ bins with each of width $b$, i.e., $\mathbb{B} = \{[0, b), [b, 2b), ..., [(T-1)b, 255]\}$. The upper bound would be 1 instead of 255 if the value range is within $[0, 1]$. Given a bin $B_m = [(t-1)b, tb) \in \mathbb{B}$ and a pixel of prediction at location $i$, we define an indicator function $\mathbf{I}_{B_m}(\mathbf{x}_{m,1:N}^{(i)})$ such that it returns 1 if $\mathbf{x}_{m,1:N}^{(i)} \in B_m$ and 0 otherwise. For multiple models, we have $M$ bins to form a bin set $(B_1, ..., B_M) \in \mathbb{B}^M$, and define the mask map as $\mathbf{R}_r = \prod_{m=1}^M \mathbf{I}_{B_m}(\mathbf{x}_{m,1:N}) \in \{0, 1\}^{NL}$ where $r = 1, ..., T^M$. It holds $\sum_{r=1}^{T^M} \mathbf{R}_r = \mathbf{1}$ and $\prod_{r=1}^{T^M} \mathbf{R}_r = \mathbf{0}$. For each bin set $(B_1, ..., B_M) \in \mathbb{B}^M$, we can select pixels within the bin set from the original image vectors by

$$\mathbf{y}_{r,1:N} = \mathbf{R}_r \cdot \mathbf{y}_{1:N}, \qquad \mathbf{x}_{r,m,1:N} = \mathbf{R}_r \cdot \mathbf{x}_{m,1:N}, \tag{3}$$

where the operation $\{\cdot\}$ denotes the element-wise product. By the central limit theorem, we assume the nonzero pixels within each bin follow a Gaussian distribution with the mean $\mu_{r,m,1:N}$ and variance $\sigma_{r,m,1:N}$, i.e.,

$$\mathbf{y}_{r,1:N}^{(i)}|f_m, \mathbf{x}_{r,m,1:N} \overset{i.i.d}{\sim} \mathcal{N}(\mu_{r,m,1:N}, \sigma_{r,m,1:N}), \; \forall i \in [1, ..., N_r], \tag{4}$$

where $N_r$ is the number of nonzero pixels in $\mathbf{R}_r$ such that $\sum_{r=1}^{T^M} N_r = NL$, and the values of $\mu_{r,m,1:N}$ and $\sigma_{r,m,1:N}$ can be estimated by the mean and variance of $N_r$ prediction pixels within the current bin set.

The reference set is therefore separated into $T^M$ number of bin sets, and the ground-truth pixels inside each of them form a solvable univariate GMM. We then introduce the latent variable $z$ such that $z = m$ if the pixel $\mathbf{y}_{r,1:N}^{(i)}$ follows the $m$-th Gaussian by the model $f_m$. It represents the probability of the pixel belonging to the $m$-th Gaussian component, which is equivalent to the role of the ensemble weight for the $m$-th base model. By writing $\alpha_{r,m} = P(z = m)$, we have

$$\mathbf{y}_{r,1:N}^{(i)} = \mathbb{E}_z \left[ \mathbf{x}_{r,m,1:N}^{(i)} \right] = \sum_{m=1}^{M} \alpha_{r,m} \cdot \mathbf{x}_{r,m,1:N}^{(i)}; \ P(\mathbf{y}_{r,1:N}^{(i)}) = \sum_{m=1}^{M} \alpha_{r,m} P \left( \mathbf{y}_{r,1:N}^{(i)} \middle| z = m \right), \ (5)$$

where $P(\mathbf{y}_{r,1:N}^{(i)} | z = m) = \phi(\mathbf{y}_{r,1:N}^{(i)}; \mu_{r,m,1:N}, \sigma_{r,m,1:N})$ is the density function of Gaussian $\mathcal{N}(\mu_{r,m,1:N}, \sigma_{r,m,1:N})$.

The value of ensemble weights can be estimated by the maximum likelihood estimates of the observed ground-truths, i.e.,

$$\{\alpha_{r,m}\}_{r,m} \in \arg\max P(\mathbf{y}_{1:N}). \tag{6}$$

Because arbitrary two bin sets are mutually exclusive, we can safely split the optimization of maximum likelihood over $\mathbf{y}_{1:N}$ into $T^M$ optimization problems of maximum likelihood over $\mathbf{y}_{r,1:N}$. Each of them is formulated as

$$\alpha_{r,m} \in \arg\max_{\alpha_{r,m}} P(\mathbf{y}_{r,1:N}) = \arg\max_{\alpha_{r,m}} \prod_{i=1}^{N_r} P(\mathbf{y}_{r,1:N}^{(i)}). \tag{7}$$

We have formulated the expression of GMMs for estimating the range-specific ensemble weights.

### 3.3 Restoration Ensemble via Expectation Maximization and Lookup Table

#### 3.3.1 Weight Estimation via EM Algorithm

For each bin set $(B_1, ..., B_M)$, we estimate ensemble weights by maximizing the log likelihood as

$$\begin{aligned}
\log P(\mathbf{y}_{r,1:N}) &= \log \prod_{i=1}^{N_r} \sum_{m=1}^{M} \alpha_{r,m} \phi \left( \mathbf{y}_{r,1:N}^{(i)}; \mu_{r,m,1:N}, \sigma_{r,m,1:N} \right) \\
&= \sum_{i=1}^{N_r} \log \sum_{m=1}^{M} \alpha_{r,m} \phi \left( \mathbf{y}_{r,1:N}^{(i)}; \mu_{r,m,1:N}, \sigma_{r,m,1:N} \right) \\
&\geq \sum_{m=1}^{M} P \left( z = m \middle| \mathbf{y}_{r,1:N}^{(i)} \right) \log \frac{\alpha_{r,m} \phi(\mathbf{y}_{r,1:N}^{(i)}; \mu_{r,m,1:N}, \sigma_{r,m,1:N})}{P \left( z = m \middle| \mathbf{y}_{r,1:N}^{(i)} \right)},
\end{aligned} \tag{8}$$

We have an E-step to estimate the posterior distribution by

$$\gamma_{r,m,1:N} \leftarrow P \left( z = m \middle| \mathbf{y}_{r,1:N}^{(i)} \right) = \frac{\alpha_{r,m} \phi \left( \mathbf{y}_{r,1:N}^{(i)}; \mu_{r,m,1:N}, \sigma_{r,m,1:N} \right)}{\sum_{m=1}^{M} \alpha_{r,m} \phi \left( \mathbf{y}_{r,1:N}^{(i)}; \mu_{r,m,1:N}, \sigma_{r,m,1:N} \right)}. \tag{9}$$

After that, we have an M-step to obtain the maximum likelihood estimates by

$$\alpha_{r,m} \leftarrow \frac{1}{N_r} \sum_{i=1}^{N_r} \gamma_{r,m,1:N} \tag{10}$$

$$\sigma_{r,m,1:N} \leftarrow \frac{\sum_{i=1}^{N_r} \gamma_{r,m,1:N} \left( \mathbf{y}_{r,m,1:N}^{(i)} - \mu_{r,m,1:N} \right)^2}{\sum_{n=1}^{N_r} \gamma_{r,m,1:N}}. \tag{11}$$

Thanks to the separation of bin sets, we have prior knowledge of the mean and variance of each model, which can be estimated and initialized by

$$\mu_{r,m,1:N} \leftarrow \frac{1}{N_r} \sum_{i=1}^{N_r} \mathbf{x}_{r,m,1:N}^{(i)} \tag{12}$$

$$\sigma_{r,m,1:N} \leftarrow \frac{1}{N_r} \left\| \mathbf{x}_{r,m,1:N} - \mu_{r,m,1:N} \right\|_2. \tag{13}$$

The complete and detailed derivation of the EM algorithm can be found in Sec. A.3 of Appendix.

### 3.3.2 Lookup Table and Inference

We store the range-specific weights estimated on the reference set into a LUT with each key of $(B_1, ..., B_M)$. During the inference stage for a test sample $\hat{\mathbf{X}}_n$, we have the prediction of the $m$-th base model as $\tilde{\mathbf{X}}_{m,n} = f_m(\hat{\mathbf{X}}_n)$. For a bin set $(B_1, ..., B_M)$, we partition input pixels of multiple models into each bin set as

$$\tilde{\mathbf{X}}_{r,m,n} = \mathbf{R}_r \cdot \tilde{\mathbf{X}}_{m,n}, \text{ where } \mathbf{R}_r = \prod_{m=1}^{M} \mathbf{I}_{B_m}(\tilde{\mathbf{X}}_{m,n}). \tag{14}$$

Then we retrieve the estimated range-wise weights $\alpha_{r,m}$ from the LUT based on each key of the bin set and obtain the aggregated ensemble by

$$\tilde{\mathbf{Y}}_n = \sum_{r=1}^{T^M} \tilde{\mathbf{Y}}_{r,n} = \sum_{r=1}^{T^M} \sum_{m=1}^{M} \alpha_{r,m} \tilde{\mathbf{X}}_{r,m,n}. \tag{15}$$

The main algorithm can be found in Algo. 1 and the EM algorithm with known mean and variance priors is shown in Algo. 2.

## 4 Experiment

### 4.1 Experimental Settings

---

**Algorithm 2: MPEM**: EM algorithm with known Mean Prior

**Input:** $\mathbf{y}_{r,1:N}, \mathbf{x}_{r,1,1:N}, ..., \mathbf{x}_{r,M,1:N}$
**Output:** $(\alpha_{r,1}, ..., \alpha_{r,M})$

1   $N_r \leftarrow$ number of nonzero pixels in $\mathbf{y}_{r,1:N}$;
2   Initialize $\mu_{r,m,1:N}$ by Eq. 12;
3   Initialize $\sigma_{r,m,1:N}$ by Eq. 13;
4   **while** *not converge* **do**
5      **for** $i \in [1, N_r]$ **do**
6          **for** $m \in [1, M]$ **do**
7              Update $\gamma_{r,m,1:N}$ by Eq. 9;
8          **end**
9      **end**
10      **for** $m \in [1, M]$ **do**
11          Update $\alpha_{r,m}$ by Eq. 10;
12          Update $\sigma_{r,m,1:N}$ by Eq. 11;
13      **end**
14 **end**

---

**Benchmarks.** We evaluate our ensemble method on 3 image restoration tasks including super-resolution, deblurring, and deraining. For super-resolution, we use Set5 [5], Set14 [83], BSDS100 [47], Urban100 [30] and Manga109 [48] as benchmarks. For deblurring, we use GoPro [51], HIDE [60], RealBlur-J and -R [58]. For deraining, we adopt Rain100H [74], Rain100L [74], Test100[88], Test1200 [86], and Test2800 [25].

**Metrics.** We use peak signal-to-noise ratio (PSNR) and structural similarity index measure (SSIM) [71] to quantitatively evaluate the image restoration quality. Additionally, we compare the average runtime per image in seconds to evaluate the ensemble efficiency. Following prior works [27, 39, 67, 81, 82, 93], PSNR and SSIM are computed on Y channel of the YCbCr color space for image super-resolution and deraining.

**Base Models.** To evaluate the generalization of ensemble methods against model choices, we employ a wide variety of base models, including CNNs, ViTs, MLPs and Mambas. For image super-resolution, we use SwinIR [39], SRFormer [93], and MambaIR [27]. We choose MPRNet [82], DGUNet [50], and Restormer [81] for deblurring, as well as MPRNet [82], MAXIM [67], and Restormer [81] for deraining.

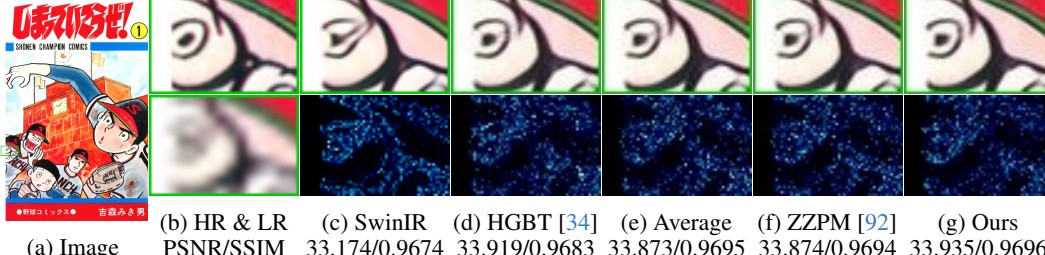

| (a) Image | (b) HR & LR PSNR/SSIM | (c) SwinIR 33.174/0.9674 | (d) HGBT [34] 33.919/0.9683 | (e) Average 33.873/0.9695 | (f) ZZPM [92] 33.874/0.9694 | (g) Ours 33.935/0.9696 |
|---|---|---|---|---|---|---|

Figure 1: A visual comparison of ensemble on an image from Manga109 [48] for the task of super-resolution. "HR & LR" means high-resolution and bicubic-upscaled low-resolution images. The second line of (c)-(g) are error maps. Please zoom in for better visual quality.

**Baselines.** We utilize regression algorithms including bagging [6], AdaBoost [22], random forests (RForest) [7], gradient boosting decision tree (GBDT) [23], histogram gradient boosting decision tree (HGBT) [14, 34] as baselines. Averaging is also a commonly used ensemble baseline. A recent method proposed by team ZZPM in the NTIRE 2023 competition [92] is also included for comparison. Additionally, we adopt RefESR [31] for image super-resolution ensemble.

**Implementation Details.** We choose the bin width as 32 by default for the balance of efficiency and performance. The EM solver of GMM stops after 1000 iterations or when the change of log likelihood is less than $1e^{-5}$. For cases where $N_r$ is fewer than 100 or the EM solution is undetermined, we use averaging weights by default. The values are reported by taking the average of 4 trials. For the construction of the reference set, we randomly select one image from the training set of DIV2K [4] for super-resolution, while for deblurring and deraining, we sample 10 images from the training sets of GoPro [51] and Rain13K [32], respectively. All the experiments are run in Python on a device with an 8-cores 2.10GHz Intel Xeon Processor and 32G Nvidia Tesla V100. The regression-based ensemble algorithms are implemented based on scikit-learn [8].

Table 1: Ablation study of bin width $b$ on Rain100H [74] with maximum step number 1000. "Runtime" is the average runtime [s].

| $b$ | 16 | 32 | 64 | 96 | 128 |
|---|---|---|---|---|---|
| Runtime | 1.2460 | 0.1709 | 0.0265 | 0.0132 | 0.0059 |
| PSNR | 31.745 | 31.739 | 31.720 | 31.713 | 31.725 |
| SSIM | 0.9093 | 0.9095 | 0.9094 | 0.9093 | 0.9093 |

Table 2: Ablation study of maximum step number in the EM algorithm on Rain100H [74] with $b = 32$. "Time" is the time of EM algorithm [s].

| #step | 10 | 100 | 500 | 1000 | 10000 |
|---|---|---|---|---|---|
| Time | 12.108 | 28.516 | 30.409 | 30.518 | 30.524 |
| PSNR | 31.734 | 31.738 | 31.738 | 31.739 | 31.739 |
| SSIM | 0.9093 | 0.9094 | 0.9095 | 0.9095 | 0.9095 |

Table 3: The ensemble results on the task of *image super-resolution*. The categories of "Base", "Regr." and "IR." in the first column mean base models, regression-based ensemble methods, and those ensemble methods designed for image restoration. The best and second best ensemble results are emphasized in **bold** and underlined respectively.

| | Datasets | Set5 [5] | | Set14 [83] | | BSDS100 [47] | | Urban100 [30] | | Manga109 [48] | |
|---|---|---|---|---|---|---|---|---|---|---|---|
| | Metrics | PSNR | SSIM | PSNR | SSIM | PSNR | SSIM | PSNR | SSIM | PSNR | SSIM |
| Base | SwinIR [39] | 32.916 | 0.9044 | 29.087 | 0.7950 | 27.919 | 0.7487 | 27.453 | 0.8254 | 32.024 | 0.9260 |
| | SRFormer [93] | 32.922 | 0.9043 | 29.090 | 0.7942 | 27.914 | 0.7489 | 27.535 | 0.8261 | 32.203 | 0.9271 |
| | MambaIR [27] | 33.045 | 0.9051 | 29.159 | 0.7958 | 27.967 | 0.7510 | 27.775 | 0.8321 | 32.308 | 0.9283 |
| Regr. | Bagging [6] | 33.006 | 0.9050 | 29.119 | 0.7950 | 27.946 | 0.7498 | 27.546 | 0.8273 | 32.154 | 0.9270 |
| | AdaBoost [22] | 33.072 | 0.9049 | 29.175 | 0.7959 | 27.975 | 0.7503 | 27.786 | 0.8302 | 32.457 | 0.9286 |
| | RForest [7] | 33.032 | 0.9052 | 29.158 | 0.7954 | 27.964 | 0.7500 | 27.640 | 0.8287 | 32.287 | 0.9279 |
| | GBDT [23] | 33.085 | 0.9050 | 29.196 | 0.7956 | 27.980 | 0.7500 | 27.792 | 0.8311 | 32.467 | 0.9285 |
| | HGBT [34] | 33.078 | 0.9051 | 29.201 | 0.7959 | 27.984 | 0.7502 | 27.783 | 0.8310 | 32.444 | 0.9282 |
| IR. | Average | 33.097 | 0.9057 | 29.202 | **0.7964** | 27.983 | 0.7506 | 27.785 | 0.8313 | 32.466 | 0.9290 |
| | RefESR [31] | 33.091 | 0.9052 | 29.172 | 0.7960 | 27.972 | 0.7504 | 27.785 | 0.8312 | 32.447 | 0.9288 |
| | ZZPM [92] | 33.094 | 0.9057 | 29.203 | 0.7963 | 27.981 | 0.7506 | 27.786 | 0.8313 | 32.467 | 0.9290 |
| | EnsIR (Ours) | **33.103** | **0.9058** | **29.205** | **0.7964** | **27.984** | **0.7507** | **27.795** | **0.8315** | **32.468** | **0.9291** |

Table 4: The ensemble results on the task of *image deblurring*. The categories of "Base", "Regr." and "IR." in the first column mean base models, regression-based ensemble methods, and those ensemble methods designed for image restoration. The best and second best ensemble results are emphasized in **bold** and underlined respectively.

| | Datasets | GoPro [51] | | HIDE [60] | | RealBlur-R [58] | | RealBlur-J [58] | |
|---|---|---|---|---|---|---|---|---|---|
| | Metrics | PSNR | SSIM | PSNR | SSIM | PSNR | SSIM | PSNR | SSIM |
| Base | MPRNet [82] | 32.658 | 0.9362 | 30.962 | 0.9188 | 33.914 | 0.9425 | 26.515 | 0.8240 |
| | Restormer [81] | 32.918 | 0.9398 | 31.221 | 0.9226 | 33.984 | 0.9463 | 26.626 | 0.8274 |
| | DGUNet [50] | 33.173 | 0.9423 | 31.404 | 0.9257 | 33.990 | 0.9430 | 26.583 | 0.8261 |
| Regr. | Bagging [6] | 33.194 | 0.9418 | 31.437 | 0.9250 | 34.033 | 0.9456 | 26.641 | 0.8277 |
| | AdaBoost [22] | 33.205 | 0.9412 | 31.449 | 0.9251 | 34.035 | 0.9455 | 26.652 | 0.8280 |
| | RForest [7] | 33.173 | 0.9416 | 31.439 | 0.9247 | 34.039 | 0.9457 | 26.647 | 0.8280 |
| | GBDT [23] | 33.311 | 0.9418 | 31.568 | 0.9256 | 34.052 | 0.9465 | 26.684 | 0.8285 |
| | HGBT [34] | 33.323 | 0.9427 | 31.583 | 0.9267 | 33.986 | 0.9436 | 26.684 | 0.8296 |
| IR. | Average | 33.330 | 0.9436 | 31.579 | 0.9277 | 34.090 | 0.9471 | 26.689 | **0.8309** |
| | ZZPM [92] | 33.332 | 0.9436 | 31.580 | 0.9277 | 34.057 | 0.9468 | 26.688 | 0.8308 |
| | EnsIR (Ours) | **33.345** | **0.9438** | **31.590** | **0.9278** | **34.089** | **0.9472** | **26.690** | **0.8309** |

Table 5: The ensemble results on the task of *image deraining*. The categories of "Base", "Regr." and "IR." in the first column mean base models, regression-based ensemble methods, and those ensemble methods designed for image restoration. The best and second best ensemble results are emphasized in **bold** and underlined respectively.

| | Datasets | Rain100H [74] | | Rain100L [74] | | Test100 [88] | | Test1200 [86] | | Test2800 [25] | |
|---|---|---|---|---|---|---|---|---|---|---|---|
| | Metrics | PSNR | SSIM | PSNR | SSIM | PSNR | SSIM | PSNR | SSIM | PSNR | SSIM |
| Base | MPRNet [82] | 30.428 | 0.8905 | 36.463 | 0.9657 | 30.292 | 0.8983 | 32.944 | 0.9175 | 33.667 | 0.9389 |
| | MAXIM [67] | 30.838 | 0.9043 | 38.152 | 0.9782 | 31.194 | 0.9239 | 32.401 | 0.9240 | 33.837 | 0.9438 |
| | Restormer [81] | 31.477 | 0.9054 | 39.080 | 0.9785 | 32.025 | 0.9237 | 33.219 | 0.9270 | 34.211 | 0.9449 |
| Regr. | Bagging [6] | 31.461 | 0.9001 | 39.060 | 0.9782 | 31.865 | 0.9107 | 33.115 | 0.9152 | 34.216 | 0.9446 |
| | AdaBoost [22] | 31.472 | 0.9006 | 39.067 | 0.9782 | 31.866 | 0.9112 | 33.117 | 0.9153 | 34.221 | 0.9443 |
| | RForest [7] | 31.492 | 0.9012 | 39.089 | 0.9784 | 31.900 | 0.9127 | 33.147 | 0.9169 | 34.224 | 0.9447 |
| | GBDT [23] | 31.581 | 0.9058 | 39.044 | 0.9778 | 32.001 | 0.9236 | 33.276 | 0.9274 | 34.211 | 0.9446 |
| | HGBT [34] | 31.698 | 0.9075 | 39.115 | 0.9784 | 31.988 | 0.9241 | 33.305 | 0.9282 | 34.229 | 0.9450 |
| IR. | Average | 31.681 | 0.9091 | 38.675 | 0.9770 | 31.626 | 0.9225 | 33.427 | 0.9286 | 34.214 | 0.9449 |
| | ZZPM [92] | 31.689 | 0.9091 | 38.725 | 0.9771 | 31.642 | 0.9227 | 33.434 | 0.9286 | 34.231 | 0.9450 |
| | EnsIR (Ours) | **31.739** | **0.9095** | **39.216** | **0.9792** | **32.064** | **0.9258** | **33.445** | **0.9289** | **34.245** | **0.9451** |

## 4.2 Experimental Results

### 4.2.1 Ablation Study

We conduct ablation studies on the bin width $b$ and the maximum step number of the EM algorithm on the benchmark of Rain100H [74]. The results are shown in Tab. 1 and Tab. 2 respectively. We can observe that the bin width involves a trade-off between ensemble accuracy and the efficiency . A small $b$ generates better ensemble results but makes ensemble inference slower. In the following experiments, we choose the bin width $b = 32$ by default for the balance of efficiency and performance. On the other hand, a large maximum step number does not necessarily yield better ensemble result. Therefore, we choose 1000 as the maximum step number.

### 4.2.2 Quantitative Results

We present quantitative comparisons with existing ensemble methods in Tab. 3 for super-resolution, Tab. 4 for deblurring, and Tab. 5 for deraining. Our method generally outperforms all existing methods across the 3 image restoration tasks and 14 benchmarks. Note that Average and ZZPM [92] generally perform better than regression-based ensemble methods. However, in cases where one of the base

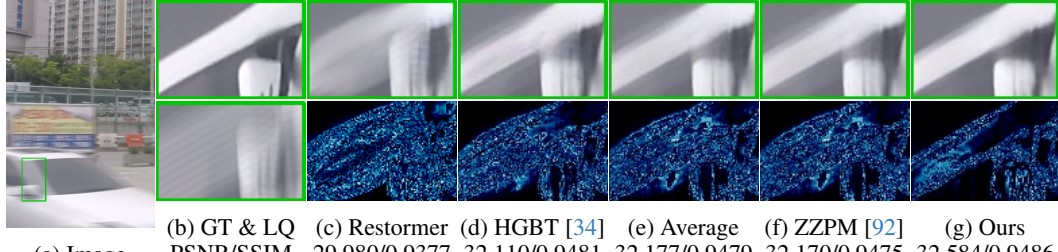

| (a) Image | (b) GT & LQ PSNR/SSIM | (c) Restormer 29.980/0.9377 | (d) HGBT [34] 32.110/0.9481 | (e) Average 32.177/0.9479 | (f) ZZPM [92] 32.170/0.9475 | (g) Ours 32.584/0.9486 |
|---|---|---|---|---|---|---|

Figure 2: A visual comparison of ensemble on an image from GoPro [51] for the task of image deblurring. "GT & LQ" means ground-truth and low quality blurry images. The second line of (c)-(g) are error maps. Please zoom in for better visual quality.

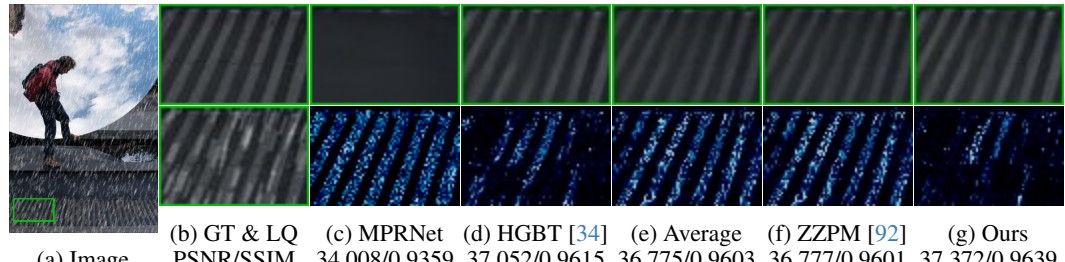

| (a) Image | (b) GT & LQ PSNR/SSIM | (c) MPRNet 34.008/0.9359 | (d) HGBT [34] 37.052/0.9615 | (e) Average 36.775/0.9603 | (f) ZZPM [92] 36.777/0.9601 | (g) Ours 37.372/0.9639 |
|---|---|---|---|---|---|---|

Figure 3: A visual comparison of ensemble on an image from Test100 [88] for the task of image deraining. "GT & LQ" means ground-truth and low quality rainy images. The second line of (c)-(g) are error maps. Please zoom in for better visual quality.

models significantly underperforms compared to the others, such as MPRNet [82] on Rain100L [74] and Test100 [88], these regression methods outperform Average and ZZPM [92]. In contrast, our method, which learns per-value weights, can recognize performance biases and alleviate such issue. ZZPM [92] performs comparably to Average in our experiments rather than outperforming it, because the base models are not always equally good and one model may be consistently better than the others. Thus, weights negatively proportional to the mean squared error may exaggerate deviation from optimal prediction. In contrast, our method consistently performs well for all cases.

### 4.2.3 Qualitative Results

We also provide qualitative visual comparisons in Fig. 1, 2 and 3. In Fig. 1, the base model SwinIR [39] mistakenly upscales the character's eye. While existing ensemble algorithms partially alleviate this mistake, they cannot fully discard the hallucinated line inside the eye. In contrast, our method with the bin width $b = 32$ that learns fine-grained range-wise weights successfully recovers the pattern. In Fig. 2, only our method can effectively obtain a better ensemble with sharp edges and accurate colors. In Fig. 3, we can observe that MPRNet [82] removes all stripe patterns on the ground together with rain streaks. The conventional weighted summation yields a dimmer ensemble result, and the HGBT method fails to learn accurate weight distributions, resulting in an unsmooth result. In contrast, ours alleviates the issue. More visual comparisons are provided in Fig. 7-18 in Appendix.

### 4.2.4 Extensions

**Efficiency.** The efficiency of ensemble methods is compared by measuring the average runtime on Rain100H [74], as shown in Tab. 6. Although our method is slower than Average and ZZPM [92],

Table 6: The average runtime per image in seconds of the ensemble methods on Rain100H [74].

| Method | Bagging [6] | AdaBoost [22] | RForest [7] | GBDT [23] | HGBT [34] | Average | ZZPM [92] | Ours |
|---|---|---|---|---|---|---|---|---|
| Runtime | 1.0070 | 1.1827 | 9.8598 | 1.2781 | 0.1773 | 0.0003 | 0.0021 | 0.1709 |

it is much faster than all the regression-based methods. By slightly sacrificing performance with $b = 64$, it can achieve real-time inference, as indicated in Tab. 1.

**Visualization.** We also present the visualization examples of ensemble weights, image features and pixel distributions in Fig. 4-6 in Appendix. Due to page limit, please refer to Sec. B of Appendix.

**Limitation and Future Work.** The trade-off between the runtime and performance has not been solved yet. Achieving real-time ensemble would lead to performance degradation. The issue could be resolved by GPU vectorization acceleration and distributed computing. Additionally, if all base models fail, ensemble methods cannot generate better result. We leave them in the future work.

## 5  Conclusion

In this paper, we propose an ensemble algorithm for image restoration based on GMMs. We partition the pixels of predictions and ground-truths into separate bins of exclusive ranges and formulate the ensemble problem using GMMs over each bin. The GMMs are solved on a reference set, and the estimated ensemble weights are stored in a lookup table for the ensemble inference on the test set. Our algorithm outperforms regression-based ensemble methods as well as commonly used averaging strategies on 14 benchmarks across 3 image restoration tasks, including super-resolution, deblurring and deraining. It is training-free, model-agnostic, and thus suitable for plug-and-play usage.

## Acknowledgement

This work has been supported in part by National Natural Science Foundation of China (No. 62322216, 62025604, 62172409), in part by Shenzhen Science and Technology Program (Grant No. JCYJ20220818102012025, KQTD20221101093559018, RCYX20221008092849068).

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

# Appendix

## A  Derivation

We first list the theorems necessary for the subsequent derivations

**Theorem A.1.** *Suppose $X \sim \mathcal{N}(\mu_X, \sigma_X)$ and $Y \sim \mathcal{N}(\mu_Y, \sigma_Y)$ are two independent random variables following univariate Gaussian distribution. If we have another random variable defined as $Z = X + Y$, then the variable follows $Z \sim \mathcal{N}(\mu_X + \mu_Y, \sigma_X + \sigma_Y)$.*

**Theorem A.2.** *Suppose $\mathbf{X} \sim \mathcal{N}(\boldsymbol{\mu}_X, \boldsymbol{\Sigma}_X)$ and $\mathbf{Y} \sim \mathcal{N}(\boldsymbol{\mu}_Y, \boldsymbol{\Sigma}_Y)$ are two independent random variables following multivariate Gaussian distribution. If we have another random variable defined as $\mathbf{Z} = \mathbf{X} + \mathbf{Y}$, then the variable follows $\mathbf{Z} \sim \mathcal{N}(\boldsymbol{\mu}_X + \boldsymbol{\mu}_Y, \boldsymbol{\Sigma}_X + \boldsymbol{\Sigma}_Y)$.*

**Theorem A.3.** *Suppose $\mathbf{X} \sim \mathcal{N}(\boldsymbol{\mu}, \boldsymbol{\Sigma})$ is a random variable following multivariate Gaussian distribution, where $\boldsymbol{\mu} \in \mathbb{R}^L$. Given a constant vector $\mathbf{a} \in \mathbb{R}^L$, we have the variable $\mathbf{Y} = \mathbf{a} \cdot \mathbf{X} \sim \mathcal{N}(\mathbf{a} \cdot \boldsymbol{\mu}, \mathbf{a}^\top \mathbf{a} \cdot \boldsymbol{\Sigma})$.*

In Sec. A.1 and A.2, it is shown the mixture of multivariate Gaussian is difficult to solve with limited samples while large feature dimension. In Sec. A.3, we derive the updating of GMMs for the pixels within a bin set. Its convergence is proved in Sec. A.4.

### A.1  Sample-wise Maximum Likelihood Estimation of multivariate Gaussian Ensemble

If we directly consider the ensemble problem by the sample-wise weighted summation of multivariate Gaussian with the ensemble weights $\beta_{n,m}$ such that $\sum_{m=1}^{M} \beta_{n,m} = 1$, i.e.,

$$\mathbf{y}_n | f_1, ..., f_M, \mathbf{x}_n \sim \mathcal{N}\left(\sum_{m=1}^{M} \beta_{n,m} \mathbf{x}_{m,n}, \sum_{m=1}^{M} \beta_{n,m}^2 \boldsymbol{\Sigma}_{m,n}\right). \tag{16}$$

The log likelihood is therefore

$$
\begin{aligned}
&\log P\left(\mathbf{y}_1, ..., \mathbf{y}_N | f_1, ..., f_M, \mathbf{x}_1, ..., \mathbf{x}_N\right) \\
&= \log \prod_{n=1}^{N} P\left(\mathbf{y}_n | f_1, ..., f_M, \mathbf{x}_n\right) \\
&= \sum_{n=1}^{N} \log \phi\left(\mathbf{y}_n; \sum_{m=1}^{M} \beta_{n,m} \mathbf{x}_{m,n}, \sum_{m=1}^{M} \beta_{n,m}^2 \boldsymbol{\Sigma}_{m,n}\right) \\
&= \sum_{n=1}^{N} -\frac{1}{2} \log\left(2\pi \sum_{m=1}^{M} \beta_{n,m}^2 \boldsymbol{\Sigma}_{m,n}\right) - \frac{\left(\mathbf{y}_n - \sum_{m=1}^{M} \beta_{n,m} \mathbf{x}_{m,n}\right)^2}{2 \sum_{m=1}^{M} \beta_{n,m}^2 \boldsymbol{\Sigma}_{m,n}}.
\end{aligned}
\tag{17}
$$

The weights can be found via computing the maximum likelihood estimates by $\frac{\partial L}{\partial \beta_{n,m}} = 0$. However, the derivative is complicated, sample-wisely different and related to both unknown and sample-wise covariance matrices. Directly making sample-wise ensemble is thereby difficult to solve.

Besides, if we use multivariate GMMs with the EM algorithm, the estimation of covariance requires the computation of its inverse. However, we only have $M$ observed samples while $L$ features with $L \gg M$ for each prediction sample. The covariance matrix is singular and thus multivariate GMMs cannot be solved.

### A.2  Set-level Maximum Likelihood Estimation of multivariate Gaussian Ensemble

If we directly consider the ensemble problem by the weighted summation of multivariate Gaussian over the reference set with the ensemble weights $\beta_{n,m}$ such that $\sum_{m=1}^{M} \beta_m = 1$, i.e.,

$$\mathbf{y}_{1:N} | f_1, ..., f_M, \mathbf{x}_{1:N} \sim \mathcal{N}\left(\sum_{m=1}^{M} \beta_m \mathbf{x}_{m,1:N}, \sum_{m=1}^{M} \beta_m^2 \operatorname{diag}(\boldsymbol{\Sigma}_{m,1}, ..., \boldsymbol{\Sigma}_{m,N})\right) \tag{18}$$

The log likelihood is therefore

$$
\begin{aligned}
&\log P(\mathbf{y}_{1:N}|f_1, ..., f_M, \mathbf{x}_{1:N}) \\
&= -\frac{1}{2}\log\left(2\pi\sum_{m=1}^{M}\beta_m^2\,\mathrm{diag}(\boldsymbol{\Sigma}_{m,1}, ..., \boldsymbol{\Sigma}_{m,N})\right) - \frac{\left(\mathbf{y}_{1:N} - \sum_{m=1}^{M}\beta_m\mathbf{x}_{m,1:N}\right)^2}{2\sum_{m=1}^{M}\beta_m^2\,\mathrm{diag}(\boldsymbol{\Sigma}_{m,1}, ..., \boldsymbol{\Sigma}_{m,N})}
\end{aligned}
\tag{19}
$$

We can compute the optimal maximum likelihood estimate by making $\frac{\partial L}{\partial \beta_m} = 0$. Because we only have $M$ observed samples while $NL$ features with $NL \gg M$ for each prediction sample. The unknown covariance matrix still cannot be estimated by multivariate GMMs with the EM algorithm,. The estimation of covariance requires the computation of its inverse. The covariance matrix is singular and thus the multivariate GMMs cannot be solved.

## A.3  Derivation of GMMs in a bin set

For each bin set, we have converted the problem into the format of GMMs, i.e.,

$$
\begin{aligned}
P(\mathbf{y}_{r,1:N}^{(i)}) &= \sum_{m=1}^{M} P(z=m)P\left(\mathbf{y}_{r,1:N}^{(i)}\middle| z=m\right) \\
&= \sum_{m=1}^{M} \alpha_{r,m}\phi\left(\mathbf{y}_{r,1:N}^{(i)}; \mu_{r,m,1:N}, \sigma_{r,m,1:N}\right),
\end{aligned}
\tag{20}
$$

where $z \in \{1, ..., M\}$ is the latent variable that represents the probability of the $m$-th mixture, and $P\left(\mathbf{y}_{r,1:N}^{(i)}\middle| z=m\right)$ is the $m$-th mixture probability component. We use $\alpha_{r,m} = P(z=m)$ to denote the mixture proportion or the probability that $\mathbf{y}_{r,1:N}^{(i)}$ belongs to the $m$-th mixture component. We assume the pixels within the bin follow the Gaussian distribution based on the central limit theorem, i.e., $\mathbf{y}_{r,1:N}^{(i)} \overset{i.i.d}{\sim} \mathcal{N}(\mu_{r,m,1:N}, \sigma_{r,m,1:N})$ and $P\left(\mathbf{y}_{r,1:N}^{(i)}\middle| z=m\right) = \phi\left(\mathbf{y}_{r,1:N}^{(i)}; \mu_{r,m,1:N}, \sigma_{r,m,1:N}\right)$. The mean and variance of the bin are denoted as $\mu_{r,m,1:N}$ and $\sigma_{r,m,1:N}$.

Suppose we have $N_r$ data samples in the bin, then for the $N_r$ observations, we have its joint probability

$$
\begin{aligned}
P(\mathbf{y}_{r,1:N}) &= P\left(\mathbf{y}_{r,1:N}^{(1)}, ..., \mathbf{y}_{r,1:N}^{(N_r)}\right) = \prod_{i=1}^{N_r} P(\mathbf{y}_{r,1:N}^{(i)}) \\
&= \prod_{i=1}^{N_r}\sum_{m=1}^{M} \alpha_{r,m}\phi\left(\mathbf{y}_{r,1:N}^{(i)}; \mu_{r,m,1:N}, \sigma_{r,m,1:N}\right)
\end{aligned}
\tag{21}
$$

Different from conventional GMMs, we have the observation prior that

$$
\mu_{r,m,1:N} = \frac{1}{N_r}\sum_{i=1}^{N_r}\mathbf{x}_{r,m,1:N}^{(i)}
\tag{22}
$$

and thus we want to find the maximum likelihood estimates of $\alpha_{r,m}$. The initial variance, $\sigma_{r,m,1:N}$ can also be estimated by

$$
\sigma_{r,m,1:N} = \frac{1}{N_r}\sum_{i=1}^{N_r}\left\|\mathbf{x}_{r,m,1:N}^{(i)} - \mu_{r,m,1:N}\right\|_2
\tag{23}
$$

**Theorem A.4.** *(Jensen's Inequality) Given a convex funnction $f$, we have $f(\mathbb{E}[X]) \geq \mathbb{E}[f(X)]$*

The log likelihood of the GMMs is

$$\log P(\mathbf{y}_{r,1:N}) = \log \prod_{i=1}^{N_r} \sum_{m=1}^{M} P(z=m) P\left(\mathbf{y}_{r,1:N}^{(i)}\middle| z=m\right)$$

$$= \sum_{i=1}^{N_r} \log \sum_{m=1}^{M} P(z=m) P\left(\mathbf{y}_{r,1:N}^{(i)}\middle| z=m\right)$$

$$= \sum_{i=1}^{N_r} \log \sum_{m=1}^{M} P\left(z=m\middle|\mathbf{y}_{r,1:N}^{(i)}\right) \frac{P(z=m) P\left(\mathbf{y}_{r,1:N}^{(i)}\middle| z=m\right)}{P\left(z=m\middle|\mathbf{y}_{r,1:N}^{(i)}\right)} \qquad (24)$$

$$= \sum_{i=1}^{N_r} \log \mathbb{E}_{z|\mathbf{y}_{r,1:N}^{(i)}} \left[ \frac{P(z=m) P\left(\mathbf{y}_{r,1:N}^{(i)}\middle| z=m\right)}{P\left(z=m\middle|\mathbf{y}_{r,1:N}^{(i)}\right)} \right]$$

$$\geq \sum_{i=1}^{N_r} \sum_{m=1}^{M} P\left(z=m\middle|\mathbf{y}_{r,1:N}^{(i)}\right) \log \frac{P(z=m) P\left(\mathbf{y}_{r,1:N}^{(i)}\middle| z=m\right)}{P\left(z=m|\mathbf{y}_{r,1:N}^{(i)}\right)},$$

Plug the expression of $P(z=m)$ and $P\left(\mathbf{y}_{r,1:N}^{(i)}\middle| z=m\right)$ in and we get

$$\log P(\mathbf{y}_{r,1:N}) \geq \sum_{i=1}^{N_r} \sum_{m=1}^{M} P(z=m|\mathbf{y}_{r,1:N}^{(i)}) \log \frac{\alpha_{r,m}\phi(\mathbf{y}_{r,1:N}^{(i)}; \mu_{r,m,1:N}, \sigma_{r,m,1:N})}{P(z=m|\mathbf{y}_{r,1:N}^{(i)})}, \qquad (25)$$

where the inequality is based on the Jensen's Inequality.

The problem can be effectively solved by the EM algorithms. We have the an E-step to estimate posterior distribution of $z$ by

$$\gamma_{r,m,1:N} \leftarrow P\left(z=m|\mathbf{y}_{r,1:N}^{(i)}\right)$$

$$= \frac{P\left(\mathbf{y}_{r,1:N}^{(i)}|z=m\right) P(z=m)}{P\left(\mathbf{y}_{r,1:N}^{(i)}\right)}$$

$$= \frac{P\left(\mathbf{y}_{r,1:N}^{(i)}|z=m\right) P(z=m)}{\sum_{m=1}^{M} P\left(\mathbf{y}_{r,1:N}^{(i)}|z=m\right) P(z=m)} \qquad (26)$$

$$= \frac{\alpha_{r,m}\phi(\mathbf{y}_{r,1:N}^{(i)}; \mu_{r,m,1:N}, \sigma_{r,m,1:N})}{\sum_{m=1}^{M} \alpha_{r,m}\phi(\mathbf{y}_{r,1:N}^{(i)}; \mu_{r,m,1:N}, \sigma_{r,m,1:N})}.$$

After that, we have an M-step to obtain maximum likelihood estimates by making the derivative of the log likelihood equal zero,

$$\alpha_{r,m} \leftarrow \frac{1}{N_r} \sum_{i}^{N_r} \gamma_{r,m,1:N} \qquad (27)$$

$$\sigma_{r,m,1:N} \leftarrow \frac{\sum_{i=1}^{N_r} \gamma_{r,m,1:N}(\mathbf{y}_{r,m,1:N}^{(i)} - \mu_{r,m,1:N})^2}{\sum_{n=1}^{N_r} \gamma_{r,m,1:N}} \qquad (28)$$

We do not update the mean because we have its prior knowledge of value as the mean of base model prediction pixels within the bin set.

### A.4 Convergence of GMMs with mean prior

Let $\theta = \{\alpha_{r,1}, ..., \alpha_{r,M}, \sigma_{r,1,1:N}, ..., \sigma_{r,M,1:N}\}$ be the estimate variable. To validate the convergence of GMMs, we want to prove

$$P\left(\mathbf{y}_{r,1:N}^{(i)}\middle|\theta^{t+1}\right) \geq P\left(\mathbf{y}_{r,1:N}^{(i)}\middle|\theta^{t}\right), \qquad (29)$$

where $\theta^t$ means the estimate variables at the $t$-th step.

We start from

$$\log P\left(\mathbf{y}^{(i)}_{r,1:N}\middle|\theta\right) = \log P\left(\mathbf{y}^{(i)}_{r,1:N}, z\middle|\theta\right) - \log P\left(z\middle|\mathbf{y}^{(i)}_{r,1:N}, \theta\right) \tag{30}$$

Based on the objective of the M-step, i.e.,

$$\theta \in \arg\max_\theta \log P\left(\mathbf{y}^{(i)}_{r,1:N}, z\middle|\theta\right). \tag{31}$$

we can naturally guarantee

$$\log P\left(\mathbf{y}^{(i)}_{r,1:N}, z\middle|\theta^{t+1}\right) \geq \log P\left(\mathbf{y}^{(i)}_{r,1:N}, z\middle|\theta^t\right). \tag{32}$$

We also have

$$\mathbb{E}_{z|\mathbf{y}^{(i)}_{r,1:N},\theta}\left[\log \frac{P\left(z\middle|\mathbf{y}^{(i)}_{r,1:N}, \theta^{t+1}\right)}{P\left(z\middle|\mathbf{y}^{(i)}_{r,1:N}, \theta^t\right)}\right] = -D_{\mathrm{KL}}\left(P\left(z\middle|\mathbf{y}^{(i)}_{r,1:N}, \theta^t\right)\middle\|P\left(z\middle|\mathbf{y}^{(i)}_{r,1:N}, \theta^{t+1}\right)\right) \leq 0, \tag{33}$$

where $D_{\mathrm{KL}}$ denotes the Kullback–Leibler (KL) divergence function.

We can thus obtain

$$\begin{aligned}
\log P\left(\mathbf{y}^{(i)}_{r,1:N}\middle|\theta^{t+1}\right) &= \log P\left(\mathbf{y}^{(i)}_{r,1:N}, z\middle|\theta^{t+1}\right) - \log P\left(z\middle|\mathbf{y}^{(i)}_{r,1:N}, \theta^{t+1}\right) \\
&\geq \log P\left(\mathbf{y}^{(i)}_{r,1:N}\middle|\theta^t\right) = \log P\left(\mathbf{y}^{(i)}_{r,1:N}, z\middle|\theta^t\right) - \log P\left(z\middle|\mathbf{y}^{(i)}_{r,1:N}, \theta^t\right).
\end{aligned} \tag{34}$$

The convergence is therefore guaranteed. The prior of mean does not affect the convergence.

## B  More Visualizations

### B.1  Weight Visualization

We plot the heatmap of ensemble weights on an example from HIDE [60]. The weight heatmaps for averaging, ZZPM [92] and ours are shown in Fig. 4. We can see that ours assigns more detailed weight on its prediction to preserve textures.

### B.2  Feature Visualization

We plot the image feature visualization via the dimension reduction of principal components analysis (PCA). We randomly sample 8 images from HIDE [60]. For each image, we obtain its restoration predictions of base models, the ensemble results of averaging, ZZPM [92], HGBT [34] and ours. A pre-trained ResNet110 is used to encode images into features. The visualizations are shown in Fig. 5 It can be found that ours can consistently yield the results closer to the gorund-truths (GT).

### B.3  Distribution Visualization

Note that we hypotheses of pixels within each bin set following Gaussian and their weights of GMMs learnt on the reference set can be used to fuse pixels on the test set. We validate them by plotting the histograms within bin sets in Fig. 6. As seen, the distribution of the pixels within a bin follows either typical Gaussian, plateau that can be considered as a flat Gaussian, or discrete signals that can be considered as steep Gaussian. The relative location of the distributions among base model predictions and ground-truths is also well preserved from the reference set to the test set.

We can notice that the simple averaging strategy can suffice for the case of the last row, while a biased averaging towards DGUNet [50] and MPRNet [82] will be better for the case of the first two rows. For the case of the third row, directly applying the prediction of DGUNet [50] will be the best. Our method estimating the range-wise weights as the latent probability of GMMs can handle all the cases, while the averaging strategies will fail.

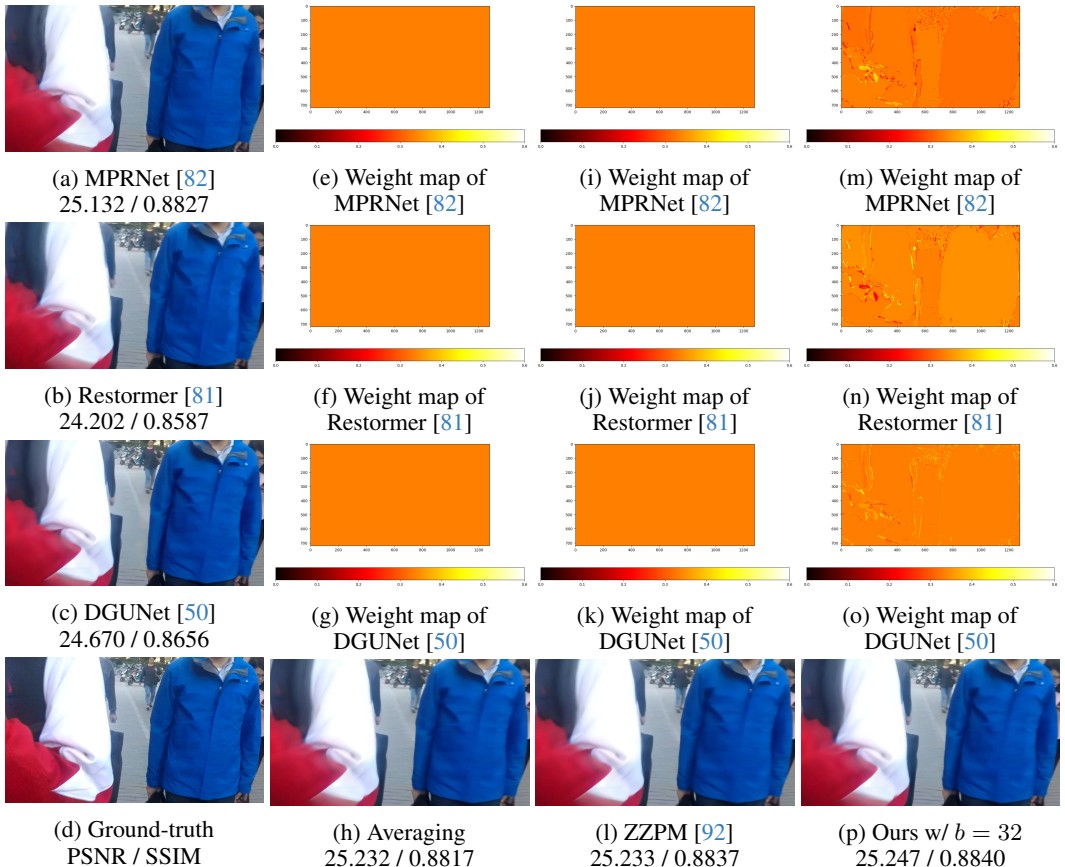

(a) MPRNet [82]
25.132 / 0.8827

(e) Weight map of
MPRNet [82]

(i) Weight map of
MPRNet [82]

(m) Weight map of
MPRNet [82]

(b) Restormer [81]
24.202 / 0.8587

(f) Weight map of
Restormer [81]

(j) Weight map of
Restormer [81]

(n) Weight map of
Restormer [81]

(c) DGUNet [50]
24.670 / 0.8656

(g) Weight map of
DGUNet [50]

(k) Weight map of
DGUNet [50]

(o) Weight map of
DGUNet [50]

(d) Ground-truth
PSNR / SSIM

(h) Averaging
25.232 / 0.8817

(l) ZZPM [92]
25.233 / 0.8837

(p) Ours w/ $b = 32$
25.247 / 0.8840

Figure 4: A sample of weight visualizations on HIDE [60]. Base models are DGUNet [50], MPR-Net [82], and Restormer [81]. The first column shows the base model predictions and ground-truth. The second column shows the ensemble weights and result of the averaging strategy. The third column shows the ensemble weights and result of ZZPM [92]. The last column shows the ensemble weights and result of our method.

## C  More Visual Comparisons

We show more visual comparisons in Fig. 7-10 for image super-resolution, in Fig. 11-13 for image deblurring, and in Fig. 14-18 for image deraining.

In Fig. 7, only ours is able to preserve gray reflection of the boat from SwinIR [39]'s prediction. In Fig. 8-10, our method obtains the most accurate ensemble results in the case that one of base models generates mistaken textures.

In Fig. 11, our method yield the sharpest and straight line like the ground-truth. In Fig. 12 and 13, our ensemble method obtains the closest textures to the ground-truths.

In Fig. 14, our method gets the best ensemble despite the mistake from MAXIM [67]. In Fig. 15, ours preserves the cleanest background to the ground-truth. In Fig. 16, only ours recovers the grid near the eyeball. In Fig. 17 and 18, ours preserves the closest background textures to the ground-truths.

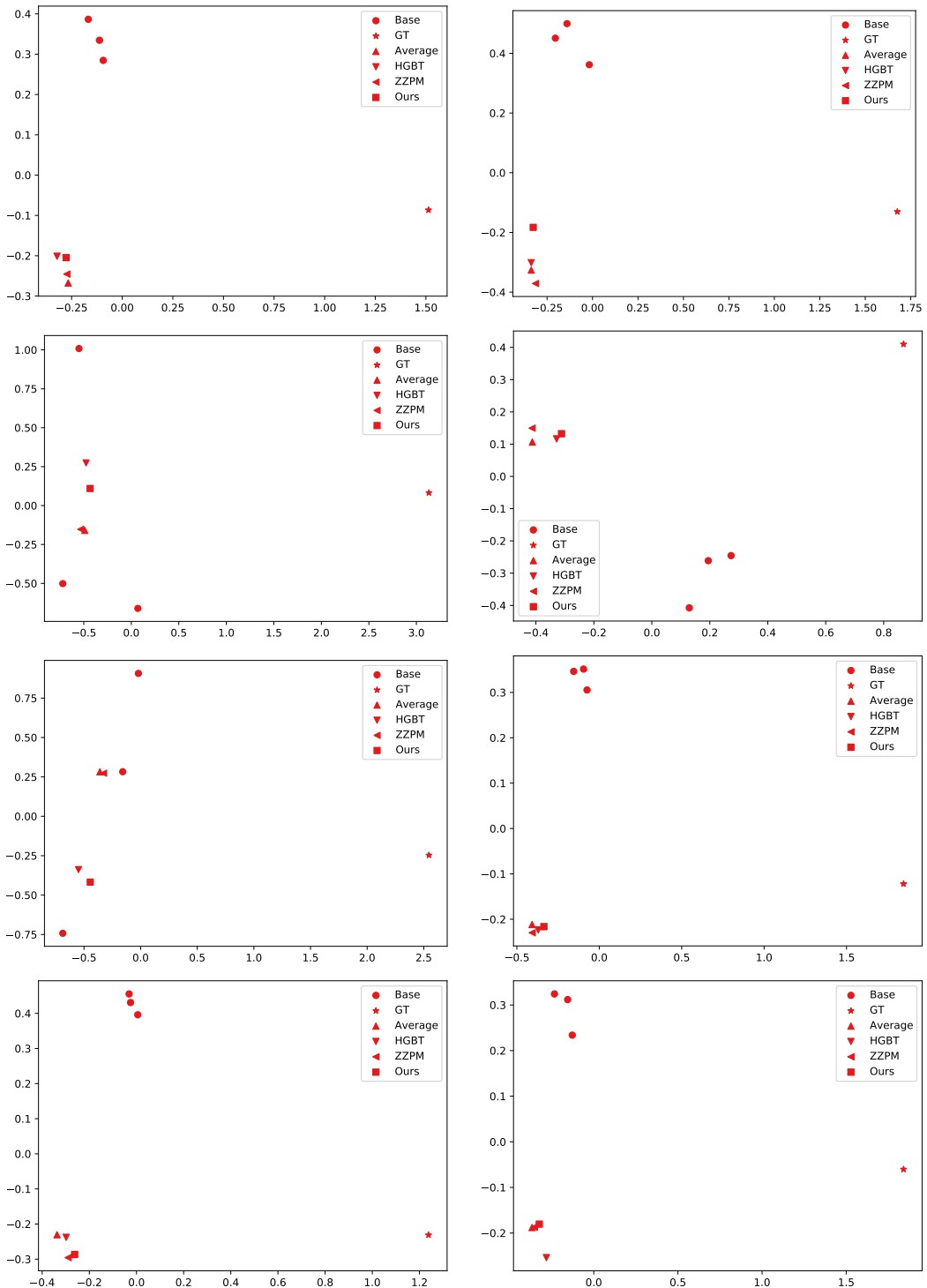

Figure 5: A sampled group of feature visualizations on HIDE [60]. "Base" denotes the features of three base models, i.e., DGUNet [50], MPRNet [82], and Restormer [81].

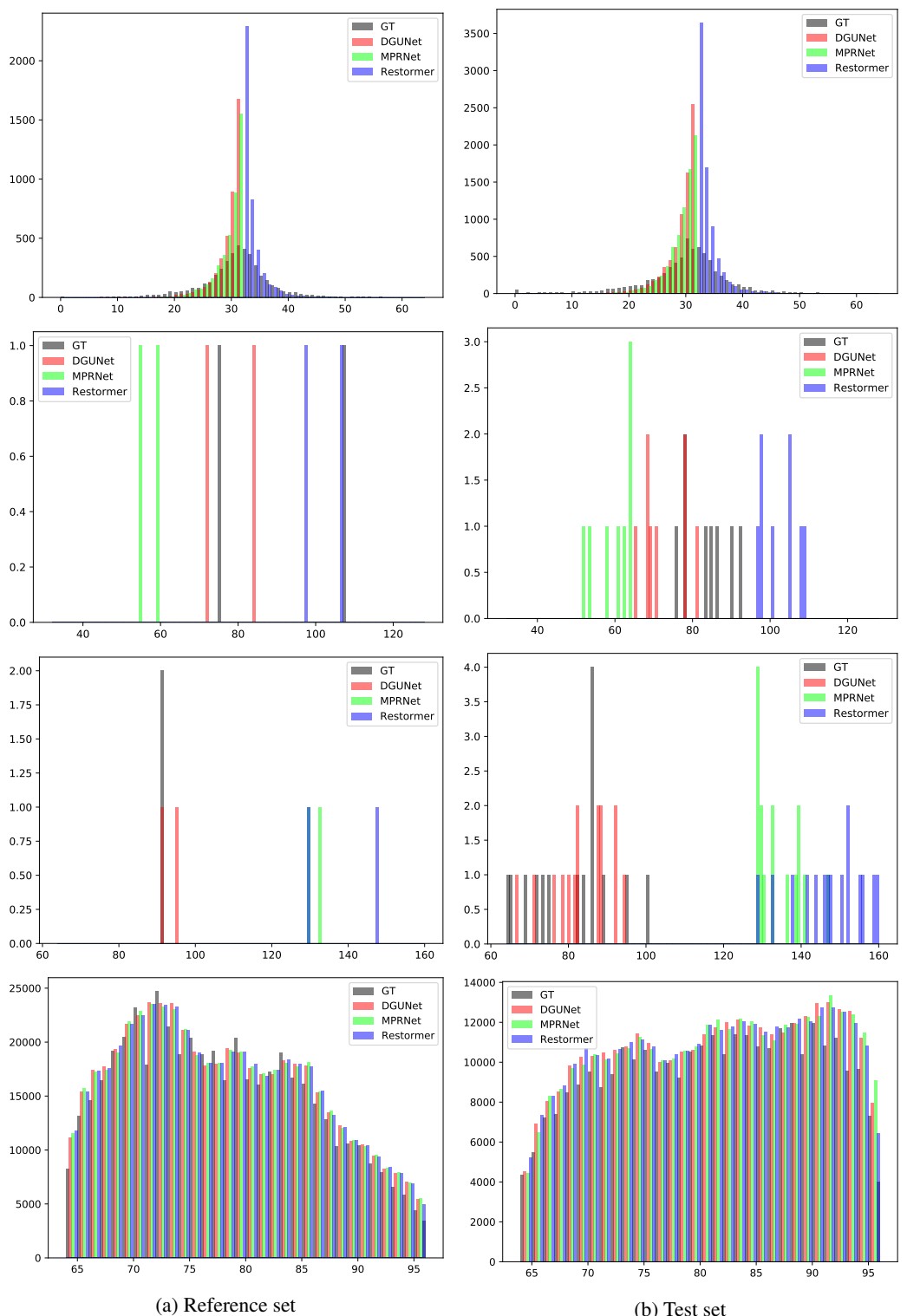

Figure 6: A group of distribution visualizations on HIDE [60]. The bin sets of the first row is $(B_1 = [0, 32), B_2 = [0, 32), B_3 = [32, 64))$. The bin sets of the second row is $(B_1 = [64, 96), B_2 = [32, 64), B_3 = [96, 128))$. The bin sets of the third row is $(B_1 = [64, 96), B_2 = [128, 160), B_3 = [128, 160))$. The bin sets of the last row is $(B_1 = [64, 96), B_2 = [64, 96), B_3 = [64, 96))$. Base models are DGUNet [50], MPRNet [82], and Restormer [81].

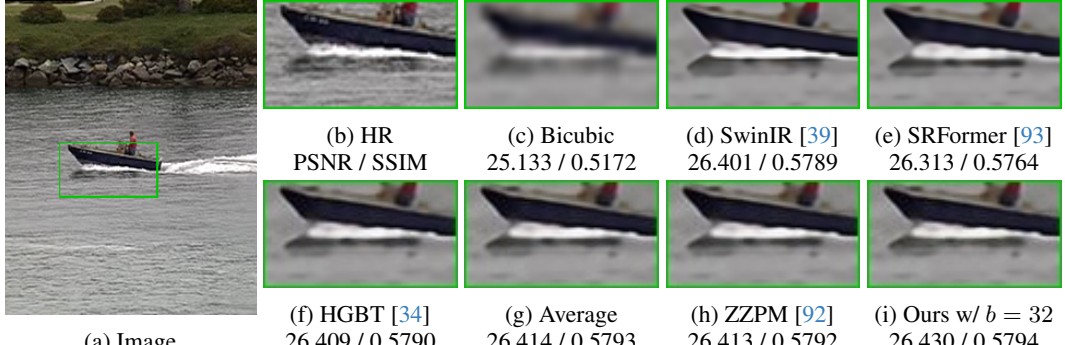

Figure 7: A visual comparison of ensemble on an image from Set14 [83] for the task of super-resolution. Please zoom in for better visual quality.

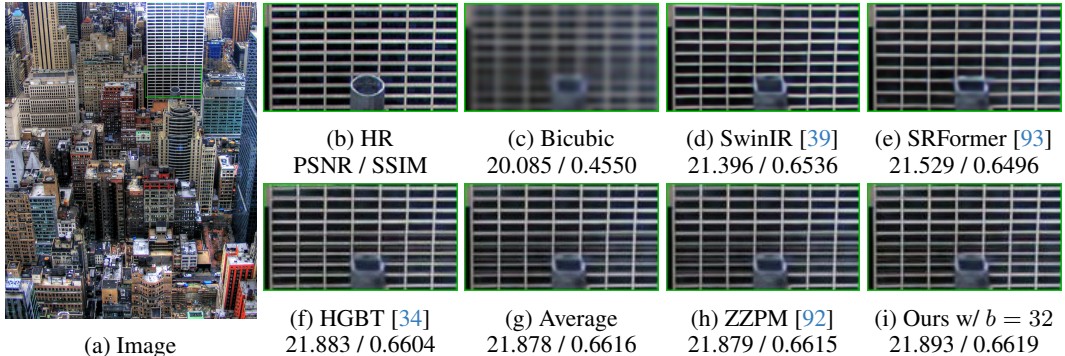

Figure 8: A visual comparison of ensemble on an image from Urban100 [30] for the task of super-resolution. Please zoom in for better visual quality.

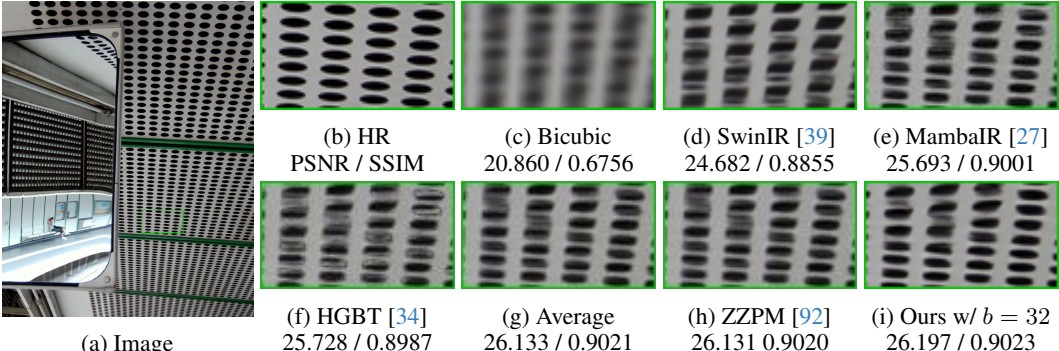

Figure 9: A visual comparison of ensemble on an image from Urban100 [30] for the task of super-resolution. Please zoom in for better visual quality.

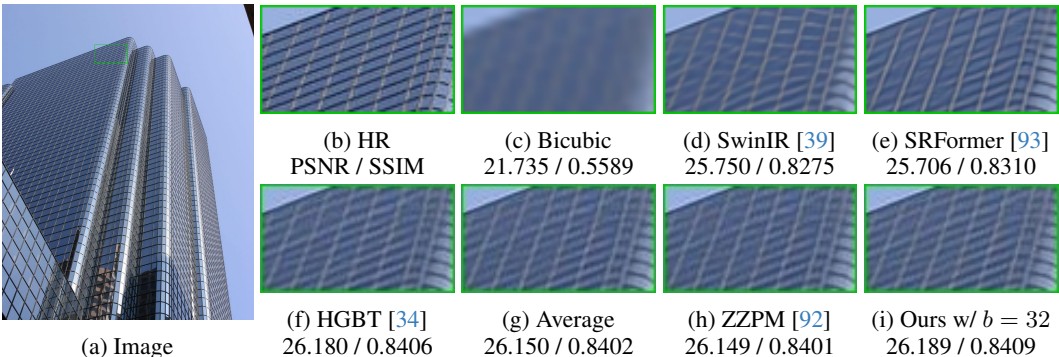

Figure 10: A visual comparison of ensemble on an image from Urban100 [30] for the task of super-resolution. Please zoom in for better visual quality.

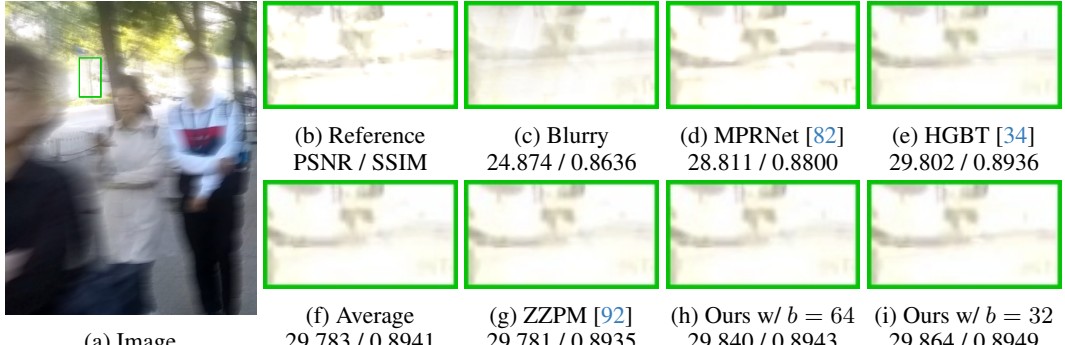

Figure 11: A visual comparison of ensemble on an image from HIDE [60] for the task of image deblurring. Please zoom in for better visual quality.

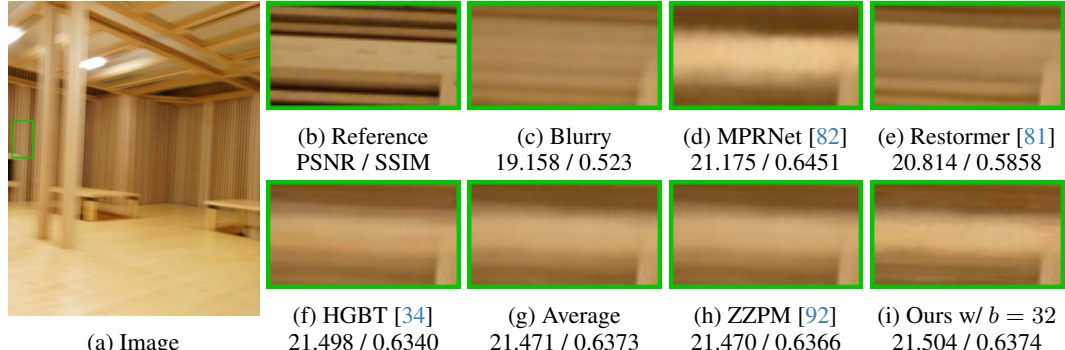

Figure 12: A visual comparison of ensemble on an image from RealBlur-J [58] for the task of image deblurring. Please zoom in for better visual quality.

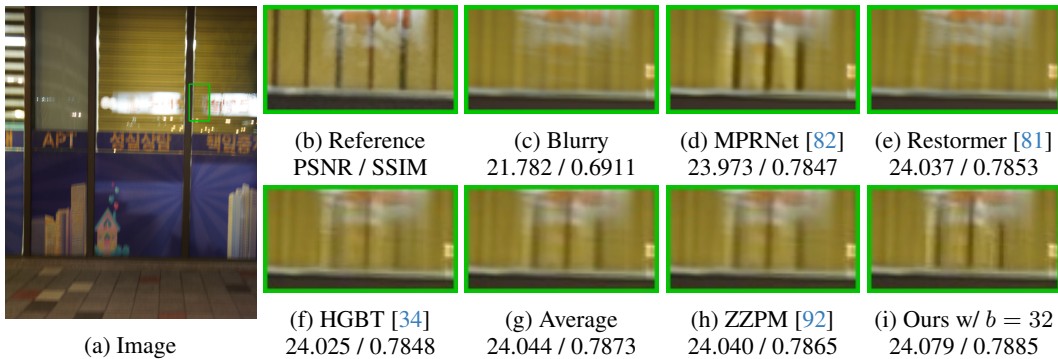

Figure 13: A visual comparison of ensemble on an image from RealBlur-J [58] for the task of image deblurring. Please zoom in for better visual quality.

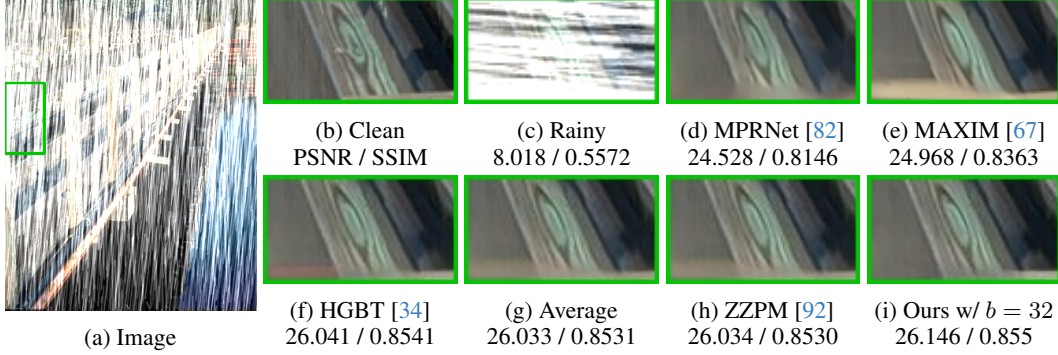

Figure 14: A visual comparison of ensemble on an image from Rain100H [74] for the task of image deraining. Please zoom in for better visual quality.

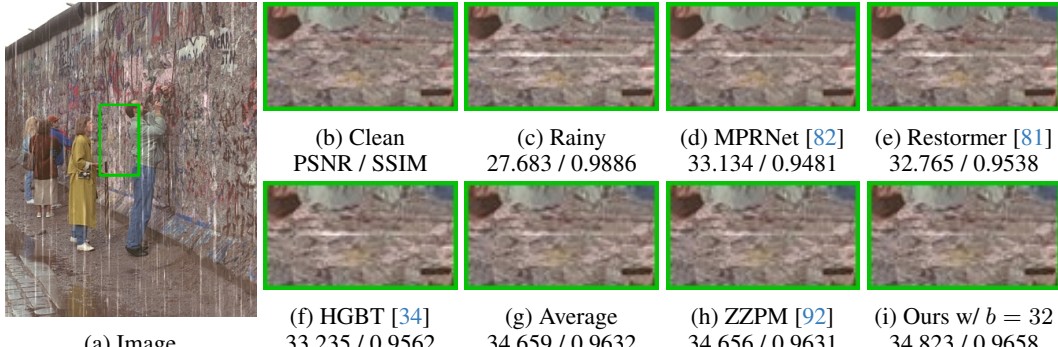

Figure 15: A visual comparison of ensemble on an image from Rain100L [74] for the task of image deraining. Please zoom in for better visual quality.

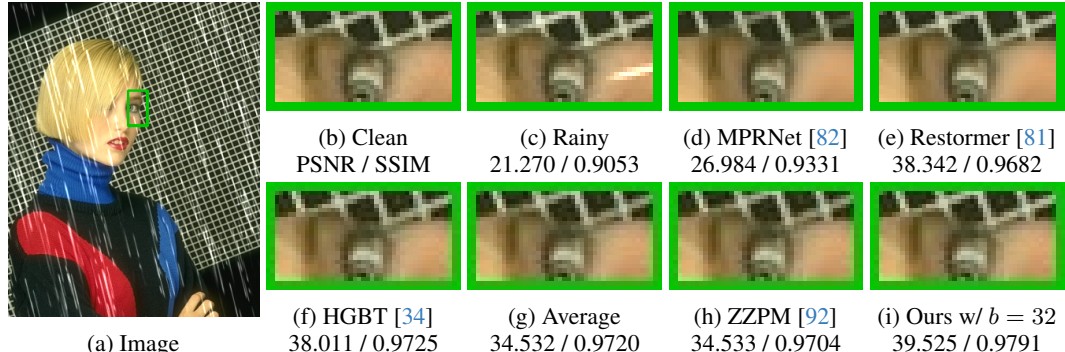

Figure 16: A visual comparison of ensemble on an image from Rain100L [74] for the task of image deraining. Please zoom in for better visual quality.

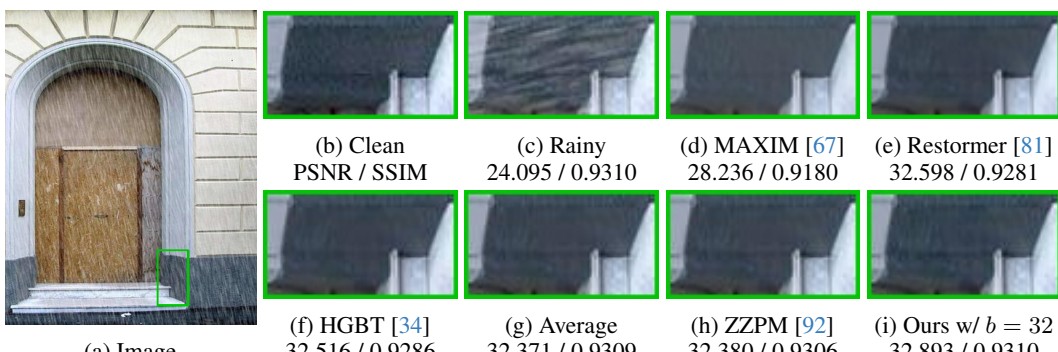

Figure 17: A visual comparison of ensemble on an image from Test1200 [86] for the task of image deraining. Please zoom in for better visual quality.

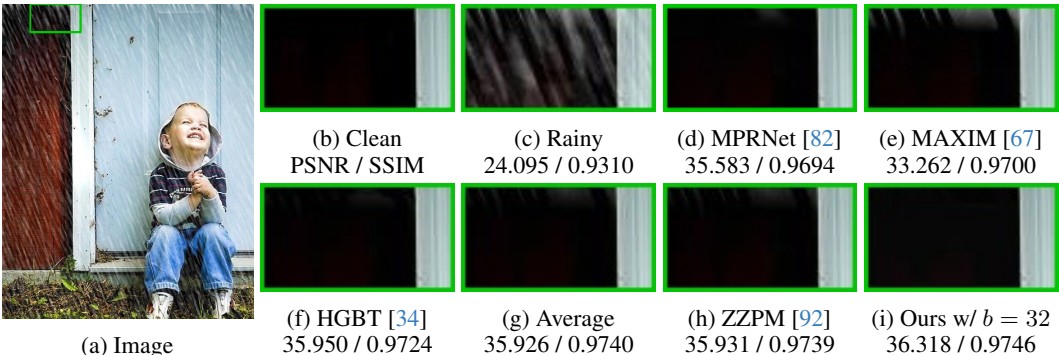

Figure 18: A visual comparison of ensemble on an image from Test2800 [86] for the task of image deraining. Please zoom in for better visual quality.

