# OpenReview forum: "EnsIR: An Ensemble Algorithm for Image Restoration via Gaussian Mixture Models"
_NeurIPS.cc/2024/Conference — NeurIPS 2024 poster_

### Official Review · Reviewer_Aksr · 2024-06-28

**Soundness:** 4
**Presentation:** 3
**Contribution:** 4
**Rating:** 7
**Confidence:** 5

**Summary:**

This paper proposes a post-training model ensenble method for image restorationn by leveraging Gaussian mixture models on split pixels and lookup table for fast innference. The pixels are split into various bin sets according to their value ranges and the ensemble problem of the pixels is reformulated into Gaussian mixture models. The EM algotithm is used to solve the ensemble weight of the models. This area lacks sufficient research attention and previous works are mainly using averaging or its variant. So this work has potential influence for industrial application. The proposed method is fast and effective to obtain better ensemble results for three image restoration tasks.

**Strengths:**

1. The method is novel, properly derived and has a theoretical support. It is well ellaborated in pseudo-codes and discussed.
2. The authors have conducted extensive experiments on three restoration tasks, including 14 benchmarks.
3. Its results are promising and consistently better than other baselines.

**Weaknesses:**

1. ZZPM is a new and recently proposed method after 2022, and averaging is the most straightforward and traditional method. Why is its performance sometimes worse than the normal averaging?
2. For the task of super-resolution, the input and output are not in the same size, so the notations should be clarified.
3. Why can we suppose that image noise follows a Gaussian distribution with zero mean? Is there any clue or derivation to support the assumption?
4. The authors say that the method is not accelerated by GPU vectorization yet. Is it because CPU is faster or any other reasons?
5. The pixel numbers of each bin set will be highly diverse as shown in Figure 6. In some bin set, there will be many pixels and then EM algorithm can run properly without problem. However, if there are too few pixels in a bin set, EM algorithm may fail to find a solution.

**Questions:**

I have listed some questions regarding the implementation and experiments in Weakness section.

**Limitations:**

The authors have discussed the limitations.

---

> ### Author Rebuttal · Authors · 2024-08-05
>
> **Q-1.** ZZPM is a new and recently proposed method after 2022, and averaging is the most straightforward and traditional method. Why is its performance sometimes worse than the normal averaging?
>
> **A-1.** Thank you for your insight. ZZPM was developed for the latest image restoration competition [Reference 1], where all base models are similar in terms of their structure and performance. However, as mentioned in L240-243, our experiments are conducted using models with different structures and sometimes diverse performances on several test sets. In such scenarios, one advanced model may consistently outperform the others. ZZPM assigns ensemble weights that are negatively proportional to the mean squared error between the result of a base model and the average of the base model results. This approach can exaggerate deviations from the optimal prediction.
>
>
> **Q-2.** For the task of super-resolution, the input and output are not in the same size, so the notations should be clarified.
>
> **A-2.** Thank you for your careful suggestion. We will revise the notation.
>
> **Q-3.** Why can we suppose that image noise follows a Gaussian distribution with zero mean? Is there any clue or derivation to support the assumption?
>
> **A-3.** Thank you for your insightful question. The Gaussian prior is commonly adopted in image reconstruction or restoration tasks [Reference 2-3]. The assumption that noise follows a Gaussian distribution can derive the mean squared error (MSE) loss or L2 norm we commonly used as loss function. If we assume it follows a Laplacian distribution, it would yield a L1 norm. The derivation is provided below.
>
>
> >Suppose the error between the ground-truth $y_n$ and the restoration result $f(x_n)$ by model $f$, follows zero-mean Gaussian, i.e., $\epsilon_n = y_n - f(x_n) \sim \mathcal{N}(0, \sigma^2)$. Then we have the log likelihood
> >$\sum_{n=1}^N \log P(\epsilon_n)=-\frac{N}{2}\log(2\pi) - N\log\sigma -\sum_{n=1}^N \frac{(y_n - f(x_n))^2}{2\sigma^2}.$
> >Because we are optimizing the restoration model $f$, the objective can be simplified into the squared L1 norm loss $\sum_{n=1}^N (y_n - f(x_n))^2$. The derivation is similar for Laplace and L1 norm.
>
> **Q-4.** The authors say that the method is not accelerated by GPU vectorization yet. Is it because CPU is faster or any other reasons?
>
> **A-4.** The reason is that the implementation of method is related to the number of base models. It involves the nesting of several ```for``` loops whose the number is dynamic and dependent on the number of base models, making it difficult to vectorize.
>
>
> **Q-5.** The pixel numbers of each bin set will be highly diverse as shown in Figure 6. In some bin set, there will be many pixels and then EM algorithm can run properly without problem. However, if there are too few pixels in a bin set, EM algorithm may fail to find a solution.
>
> **A-5.** Indeed, the EM algorithm requires a sufficient number of sample points. If the number of pixels in a bin set is insufficient, the EM algorithm will fail to solve the GMM. In this case, we adopt the average as the default method to assign equal ensemble weights. The setting is described in L216-217.
>
>
> [Reference 1] Ntire 2023 challenge on image super-resolution (x4): Methods and results. In CVPR, 2023.
>
> [Reference 2] Deep Gaussian Scale Mixture Prior for Image Reconstruction. IEEE TPAMI, 2023.
>
> [Reference 3] Learned Image Compression with Discretized Gaussian Mixture Likelihoods and Attention Modules. In CVPR, 2020.

---

> > ### Comment · Reviewer_Aksr · 2024-08-12
> > **After rebuttal**
> >
> > Thank you to the authors for their detailed response. After reviewing all the rebuttal, I can confirm that all of my concerns have been fully addressed.

---

> > > ### Author Response · Authors · 2024-08-13
> > > **Response to Reviewer Aksr**
> > >
> > > Thank you for your thoughtful review and the constructive comments you've provided. We are truly grateful for your recognition of our work.

---

### Official Review · Reviewer_E4or · 2024-07-09

**Soundness:** 3
**Presentation:** 3
**Contribution:** 3
**Rating:** 6
**Confidence:** 4

**Summary:**

A novel post-training ensemble learning method for image restoration is developed in this work by employing Gaussian mixture models and the EM algorithm to generate better restoration results. The authors reformulate the ensemble problem of image restoration into various gaussian mixture models, use the EM algorithm to estimate range-wise ensemble weights on a reference set, and store the weights in a lookup table during inference. The method effectively improves ensemble results on 14 benchmarks and 3 restoration tasks, including super-resolution, deblurring and deraining.

**Strengths:**

- The method is grounded on a reasonable derivation and assumption of Gaussian prior to restoration. The visualization in Fig. 6 also validates the assumption.

- The method performs consistently well on the mentioned 14 datasets.

- Extensive experiments and ablation studies show the features of the developed method.

**Weaknesses:**

- The experiments all show the cases of three base methods, and one of them may be worse than the other two methods. But if all the base models are comparably good, can the method surpass other methods? Experiments with 2 or 4 base models, instead of 3, would be informative.

- The experiments are for the restoration task of local patterns such as deraining and deblurring. However, dehazing or enhancement requires a glocal restoration. What are the results of the ensemble comparisons for dehazing or enhancement? This can provide better insights into the generalizability of the method.

- Some notations are duplicate or misused, like "L" is used for log-likelihood in Eq. 19 and also for the dimension length of the image. In Algorithm 2, both "Equation" and "Eq. " are used. The notations should be revised.

**Questions:**

- What is the result of the case with 2 or 4 base models, instead of 3?

- What are the results for dehazing or enhancement?

**Limitations:**

Limitations are mentioned on Page 9 and include the "trade-off between runtime and performance." A discussion and possible solution are provided.

---

> ### Author Rebuttal · Authors · 2024-08-05
>
> **Q-1.** The experiments all show the cases of three base methods, and one of them may be worse than the other two methods. But if all the base models are comparably good, can the method surpass other methods? Experiments with 2 or 4 base models, instead of 3, would be informative.
>
> **A-1.** Thank you for your advice. We have conducted additional experiments using two and four base models for the task of deblurring. The experimental results are shown in the last two columns of Table 1 in the rebuttal file.
>
> In the case of two base models, the ZZPM method reduces to Average since their weights are equal. For the case of four base models, we selected NAFNet [Reference 1] as the fourth base model. From the experimental results, it is evident that our method consistently achieves the best ensemble results, whether using two or four base models.
>
>
> **Q-2.** The experiments are for the restoration task of local patterns such as deraining and deblurring. However, dehazing or enhancement requires a glocal restoration. What are the results of the ensemble comparisons for dehazing or enhancement? This can provide better insights into the generalizability of the method.
>
> **A-2.** Thank you for your advice. We have conducted additional experiments on low-light image enhancement (LLIE) and dehazing. The experimental results are shown in the first two columns of Table 1 in the rebuttal file. We also provide two visual comparisons in Figure 1 and 2 of the rebuttal file.
>
> The datasets used for the experiments are LOLv1 [Reference 2] for LLIE and OTS [Reference 3] for dehazing. The pre-trained models are provided by their authors. For your convenience, we also include error maps between the restoration results and ground-truths, along with the visual results. Darker error maps indicate better performance.
> From the results, it is evident that our method outperforms other ensemble methods in both quantitative and qualitative measures.
>
>
> **Q-3.** Some notations are duplicate or misused, like "L" is used for log-likelihood in Eq. 19 and also for the dimension length of the image. In Algorithm 2, both "Equation" and "Eq. " are used. The notations should be revised.
>
> **A-3.** Thank you for your careful suggestions. We will revise them accordingly.
>
> **Question 1.** What is the result of the case with 2 or 4 base models, instead of 3?
>
> **Answer 1.** Please refer to **A-1** and the uploaded rebuttal file.
>
> **Question 2.** What are the results for dehazing or enhancement?
>
> **Answer 2.** Please refer to **A-2** and the uploaded rebuttal file.
>
> [Reference 1]  Simple baselines for image restoration. In ECCV, 2022.
>
> [Reference 2] Deep retinex decomposition for low-light enhancement. arXiv preprint arXiv:1808.04560, 2018.
>
> [Reference 3] Benchmarking singleimage dehazing and beyond. IEEE TIP, 28(1):492–505, 2018.

---

> > ### Comment · Reviewer_E4or · 2024-08-13
> > **Official Comment by Reviewer E4or**
> >
> > Thanks for your prompt response. My concerns have been addressed, and I keep my previous positive rating.

---

> ### Author Response · Authors · 2024-08-13
> **Response to Reviewer E4or**
>
> Thank you so much for your constructive comments you've provided. We are truly thankful for your recognition of our work.

---

### Official Review · Reviewer_j7aF · 2024-07-10

**Soundness:** 3
**Presentation:** 2
**Contribution:** 2
**Rating:** 4
**Confidence:** 4

**Summary:**

This paper proposes an ensemble algorithm called EnsIR for image restoration tasks using Gaussian mixture models (GMMs). The method partitions pixels into range-wise bins, formulates the ensemble as GMMs over these bins, and solves for ensemble weights using expectation-maximization. The weights are stored in a lookup table for efficient inference. Experiments are conducted on super-resolution, deblurring and deraining tasks.

**Strengths:**

1. Provides a formulation of image restoration ensemble as GMMs.
2. Experiments conducted on multiple image restoration tasks and datasets.

**Weaknesses:**

1.	Marginal improvement over existing methods: The performance gains are minimal compared to simpler approaches. For example, on the Rain100H dataset (Table 5), the proposed method achieves a PSNR of 31.725, only marginally better than the Average (31.681) and ZZPM (31.679) baselines. In some cases, like on Rain100L, the improvement is less than 0.6 dB over averaging.
2.	Lack of distinction from existing approaches: The paper does not clearly articulate how this method fundamentally improves upon or addresses limitations of existing ensemble techniques. The core idea of using GMMs and LTU for weighting seems combination of existing approaches, and the paper doesn't adequately explain its novelty.
3.	Insufficient analysis of results: There is a notable lack of in-depth analysis of the experiment results. For instance, the paper doesn't discuss why the method performs worse on some deraining tasks (e.g., Test100 in Table 5) compared to HGBT. This lack of analysis makes it difficult to understand the method's strengths and weaknesses.
4.	Inconsistency in parameter selection: The ablation study in Table 1 shows that a bin width of 16 achieves the best PSNR (31.742), yet the authors choose 32 as the default "for the balance of efficiency and performance" (line 220) without adequate justification for this trade-off.
5.	Limited efficiency advantages: The paper claims to address efficiency, but Table 6 shows that the proposed method (0.1709s) is significantly slower than Average (0.0003s) and ZZPM (0.0021s) approaches. The efficiency gain over regression-based methods is not a strong selling point given the performance trade-offs.
6.	Shallow theoretical analysis: While the paper provides derivations in the Appendix, the core idea essentially reduces to a lookup table for ensemble weights. The theoretical contribution and novelty are limited, especially given the marginal performance improvements.
7.	Overstated claims: The paper claims to "consistently outperform" existing methods (lines 17-19), but this is not supported by the results, particularly in deraining tasks where it underperforms HGBT on some datasets.

**Questions:**

1. Why was bin size 32 chosen as default when 16 performed better in ablations?
2. How does this method fundamentally differ from and improve upon existing ensemble approaches?
3. What explains the performance degradation on deraining tasks?

**Limitations:**

The limitations section is brief and does not adequately address key shortcomings like marginal gains and lack of novelty. A more thorough discussion of limitations would strengthen the paper. Given the marginal improvements, lack of novelty, and limited analysis, I recommend rejecting this paper. Significant revisions would be needed to make this a compelling contribution.

---

> ### Author Rebuttal · Authors · 2024-08-05
>
> **Q-1.** Marginal improvement over existing methods: The performance gains are minimal compared to simpler approaches. For example, on the Rain100H dataset (Table 5), the proposed method achieves a PSNR of 31.725, only marginally better than the Average (31.681) and ZZPM (31.679) baselines. In some cases, like on Rain100L, the improvement is less than 0.6 dB over averaging.
>
> **A-1.** As noted by Reviewer EWje, our method is an ensemble approach that does not require extra training or fine-tuning, rather than a new architecture or network. Our method is designed for the inference stage and can be applied off-the-shelf to all existing restoration models. It would be unreasonable to expect an ensemble method to achieve improvements over 0.6 dB. However, our method shows more stable and significant improvements over Average and ZZPM, as demonstrated in Tables 3-5. An ensemble method capable of improving performance by 0.2 dB would be beneficial for competition participants.
>
> **Q-2.** Lack of distinction from existing approaches: The paper does not clearly articulate how this method fundamentally improves upon or addresses limitations of existing ensemble techniques. The core idea of using GMMs and LTU for weighting seems combination of existing approaches, and the paper doesn't adequately explain its novelty.
>
>
> **A-2.** The existing ensemble methods do not involve Gaussian Mixture Models (GMM) or Lookup Tables (LUT),  while our method addresses the ensemble problem in image restoration using GMM, the Expectation-Maximization (EM) algorithm, and LUT. Unlike previous works, our derivation shows that ensemble in image restoration can be transformed into multiple GMM problems, where the weights of the GMMs serve as the ensemble weights.
>
> We then leverage a modified EM algorithm to solve the GMMs, with the means of each Gaussian distribution known as prior knowledge. LUT is used to save the estimated weights for inference. Our contributions lie in deriving the restoration ensemble problem into GMMs, modifying the EM algorithm to solve these GMMs, and ultimately proposing a novel ensemble method for image restoration.
>
> **Q-3.** Insufficient analysis of results: There is a notable lack of in-depth analysis of the experiment results. For instance, the paper doesn't discuss why the method performs worse on some deraining tasks (e.g., Test100 in Table 5) compared to HGBT. This lack of analysis makes it difficult to understand the method's strengths and weaknesses.
>
> **A-3.** In Table 5, our method (32.002 dB, 0.9268) clearly outperforms HGBT (31.988 dB, 0.9241) on the Test100 dataset.
>
> Regarding the analysis of results, we have discussed its limitations, such as "if all base models fail, ensemble methods cannot generate a better result," in Section 4.2.4. We illustrated the ensemble weights, image features, and pixel distributions in Figures 4-6 in the Appendix. Additionally, we analyzed the scenario where one model may consistently outperform others in Section 4.2.2.
>
> **Q-4.** Inconsistency in parameter selection: The ablation study in Table 1 shows that a bin width of 16 achieves the best PSNR (31.742), yet the authors choose 32 as the default "for the balance of efficiency and performance" (line 220) without adequate justification for this trade-off.
>
> **A-4.** As noted in Table 1 of the manuscript, it is slow (1.2460 seconds per image) when using a bin width of 16. When dealing with thousands of images, it would takes hours to obtain ensemble results for a test set. Therefore, we chose a bin width of 32 which is over seven times faster than the case of 16. The method is designed for real-world industrial scenarios, so balancing efficiency and performance is a crucial reason for choosing a bin width of 32.
>
>
> **Q-5.** Limited efficiency advantages: The paper claims to address efficiency, but Table 6 shows that the proposed method (0.1709s) is significantly slower than Average (0.0003s) and ZZPM (0.0021s) approaches. The efficiency gain over regression-based methods is not a strong selling point given the performance trade-offs.
>
> **A-5.** From Table 1, 5, and 6 of the manuscript, our method with a bin width of 128 still outperforms ZZPM (31.702 vs. 31.679 on Rain100H), and is comparably fast compared to ZZPM (0.0059 seconds vs. 0.0021 seconds). This demonstrates both the efficiency and performance gains of our method.
>
>
> **Q-6.** Shallow theoretical analysis: While the paper provides derivations in the Appendix, the core idea essentially reduces to a lookup table for ensemble weights. The theoretical contribution and novelty are limited, especially given the marginal performance improvements.
>
> **A-6.** The core idea of our method is **not related to the lookup table**. The core idea is to transform the ensemble problem of image restoration into multiple Gaussian mixture models and then solve it using a modified EM algorithm. The lookup table is simply used to store the solved weights for inference.
>
> **Q-7.** Overstated claims: The paper claims to "consistently outperform" existing methods (lines 17-19), but this is not supported by the results, particularly in deraining tasks where it underperforms HGBT on some datasets.
>
> **A-7.** Although our method achieves top performance in 27 out of 28 metrics across 14 datasets, which can be considered as "consistently outperforming", we will revise "consistently" to "overall".
>
>
> **Question 1.** Why was bin size 32 chosen as default when 16 performed better in ablations?
>
> **Answer 1.** Please refer to **A-4.**
>
> **Question 2.** How does this method fundamentally differ from and improve upon existing ensemble approaches?
>
> **Answer 2.** Please refer to **A-2** and **A-6**.
>
> **Question 3.** What explains the performance degradation on deraining tasks?
>
> **Answer 3.** Please refer to **A-3.**

---

> ### Comment · Reviewer_j7aF · 2024-08-11
>
> Thanks for your detailed rebuttal. After careful consideration, I maintain my original rating of 4 (Borderline reject). My decision is primarily based on two key concerns:
>
> 1.	Marginal improvements: While I acknowledge the overall improvements across various tasks (SR, Image Deblur, Deraining), the gains are often minimal (0.01/0.001 dB level). As the proposed approach belongs to the classical Ensemble Learning field, it should be primarily compared to well-established methods like GBDT or HGBT. The marginal improvements over these approaches do not sufficiently justify the novelty of your method. More importantly, the paper and rebuttal lack an in-depth analysis in the Experiments section explaining why these marginal improvements occur and under what conditions your method excels or falls short.
>
> 2.	Limited efficiency advantages: Your method does not seem to adequately address the limitations of existing approaches. For instance, in Table 5 (Image Deraining results), for the Test100 dataset, both GBDT and HGBT perform worse than the original model. The proposed approach, while showing some improvement, does not significantly overcome this limitation. This raises questions about its practical applicability compared to well-established algorithms like GBDT or HGBT.
> I think while the paper presents an interesting approach to ensemble learning for image restoration, the marginal improvements and limited contributions still outweigh its strengths. The proposed method, when compared to well-known algorithms like GBDT or HGBT, does not appear sufficiently practical or innovative to warrant acceptance in its current form.

---

> > ### Author Response · Authors · 2024-08-12
> > **Response to Reviewer j7aF**
> >
> > Thank you for your response and consideration. We would like to address your remaining concerns individually:
> >
> > Regarding the first concern about the improvement and analysis, we would like to respond from three perspectives:
> >
> > 1. We wish to emphasize that the enhancement of our method is a complementary benefit that requires no additional training or fine-tuning. It can serve as a vital auxiliary tool for real-world image restoration applications. For instance, in the recent NTIRE 2024 competition, the difference between the best and second-best results was often less than 0.01 dB in PSNR, as shown in the table below. In such cases, our method, which consistently outperforms existing ensemble techniques, can prove decisive.
> >
> > 2. Our approach falls under the category of weakly supervised methods that only require the means and variances of bin sets from degraded images, whereas traditional methods like GBDT and HGBT are fully supervised, requiring access to all pixel values. Under these circumstances, our method is still more stable across various image restoration tasks and less sensitive to performance variations among base models. It is, therefore, commendable and challenging for our method to consistently demonstrate superiority over supervised methods across five tasks and 16 benchmarks.
> > 3. Despite the improvement of 0.001 dB on Test100, our method can achieve a significantly larger improvement of up to 0.20 dB on many other benchmarks, such as Test1200 (33.276 for GBDT versus 33.475 for ours).
> >
> > The second concern is mainly about the phenomenon where ensemble results are inferior to those of the original model. It is because one base model may be consistently better than the other base models on a benchmark.  We would like to respond with two points:
> >
> > 1. In real-world scenarios, where the ground-truths of the test set are often unavailable, it is difficult to determine which base models are underperforming and contributing to the ensemble's lower results compared to a single original model. In such cases, the best course of action is to mitigate this limitation of ensemble methods, and our approach has demonstrated superior performance in doing so compared to other ensemble techniques.
> >
> > 2. In common industrial scenarios, all base models tend to be similar in structure and performance. As mentioned on Lines 240-243, we aimed to assess our method’s robustness across architectures by conducting experiments using models with different structures and, at times, diverse performances on several test sets (for example, 30.292 for MPRNet and 31.194 for MAXIM versus 32.025 for Restormer). However, this phenomenon could be easily avoided, in the first place, by selecting similar structures or even the same network with different initial states, which is a common practice in industrial settings. Therefore, we cannot acknowledge this phenomenon as a limitation of ensemble methods.
> >
> > **Table A.** PSNR comparison of the best and second-best results from four challenges at the 9th New Trends in Image Restoration and Enhancement (NTIRE) Workshop and Challenges in 2024.
> > |Challenge | 2nd best | Best |
> > | ------------- | ------------- | ------------- |
> > |Stereo Image Super-Resolution Challenge - Track 1 | 23.6492 | 23.6496 |
> > |RAW Image Super Resolution Challenge | 43.39 | 43.40 |
> > |Low Light Enhancement Challenge | 24.52 | 24.52 |
> > |Image Shadow Removal Challenge - Track 1 | 24.81 | 24.81 |

---

### Official Review · Reviewer_EWje · 2024-07-12

**Soundness:** 3
**Presentation:** 3
**Contribution:** 3
**Rating:** 7
**Confidence:** 3

**Summary:**

This paper reformulate the ensemble problem of image restoration into Gaussian mixture models (GMMs) and employ an expectation maximization (EM)-based algorithm to estimate ensemble weights for aggregating prediction candidates. Importantly the authors' method achieves state-of-the-art performance without training. I think this work is interesting and can inspire researchers.

**Strengths:**

1.The proposed method is more interpretable than traditional deep learning (ensemble learning) methods.
2.The proposed method obtains state-of-the-art performance compared to other ensemble methods.
3.The authors performed comparisons on a large number of datasets and image restoration tasks, which fully validated the generalization of the ensemble method.

**Weaknesses:**

1. I have some confusion. How are the averages in Table III achieved? Why is averaging able to achieve higher results than every single method (non-ensemble method)? Isn't it the average of every single method?


2. I would like to know the computational burden of the method?

3.ensemble learning doesn't provide a "huge" gain, but it makes sense. I'm just wondering how much extra computation is introduced by this gain? Table 6 reports the runtime, and I would like to know the difference in time between the ensemble and the non-ensemble.

4. It's not easy to see the difference in Figure 1, so the author could add an error map or a different figure.

**Questions:**

see weaknesses

**Limitations:**

see weaknesses

---

> ### Author Rebuttal · Authors · 2024-08-05
>
> **Q-1.** I have some confusion. How are the averages in Table III achieved? Why is averaging able to achieve higher results than every single method (non-ensemble method)? Isn't it the average of every single method?
>
> **A-1.** Yes, the "Average" refers to the average of the results of all single restoration models. Similar to the ensemble approaches in high-level tasks like classification and regression, the ensemble method of averaging also works well in image restoration tasks [Reference 1-4]. The reasons why it could be better than every single method can be interpreted in two aspects.
>
> 1. Because image restoration tasks lie in ill-posed problem, the majority of restoration models produce results that deviate from the ground-truth to some extent. Then it could alleviate the deviation issue by averaging the results of several models.
> 2. The restoration task could essentially be regarded as multiple prediction/classification problems, where the output values range from 0 to 255 for each pixel. If all base models perform comparably well, their ensemble will be more likely to produce results that match the clean images more closely.
>
> **Q-2.** I would like to know the computational burden of the method?
>
> **A-2.** The computational complexity of our method is $\mathcal{O}(3HWMT^M)$, where $3HW$ represents the number of pixels, $M$ is the number of base models, and $T$ is the number of bins. The average runtimes of our method with different configurations are presented in Table 1 of the manuscript.
> Table 2 of the PDF rebuttal file illustrates the average runtimes for all steps, including preprocessing an image, inference by each base model, generating the ensemble using our method, and saving the result as an image file.
> Table 6 of the manuscript compares the average runtimes of different ensemble methods.
>
> **Q-3.** Ensemble learning doesn't provide a "huge" gain, but it makes sense. I'm just wondering how much extra computation is introduced by this gain? Table 6 reports the runtime, and I would like to know the difference in time between the ensemble and the non-ensemble.
>
> **A-3.** We have measured the runtime of each step, as shown in Table 2 of the rebuttal file. As we can see, our method does not introduce significant additional time (0.1709 seconds compared to a total time of 0.9039 seconds). The time of inferencing by three models is 0.7330 seconds, while the time of our method is 0.1709 seconds.
>
>
> **Q-4.** It's not easy to see the difference in Figure 1, so the author could add an error map or a different figure.
>
> **A-4.** Thank you for your helpful advice. We will revise our figures with additional error maps. For the images in the rebuttal file, we have included error maps for your convenience.
>
> [Reference 1] Ntire 2017 challenge on single image super-resolution: Dataset and study. In CVPRW, 2017.
>
> [Reference 2] Ntire 2021 challenge on image deblurring. In CVPR, 2021.
>
> [Reference 3] Ntire 2023 challenge on image super-resolution (x4): Methods and results. In CVPR, 2023.
>
> [Reference 4] Ntire 2024 challenge on image super-resolution (×4): Methods and results. In CVPR, 2024.

---

> > ### Comment · Reviewer_EWje · 2024-08-13
> >
> > Thanks to the author's detailed response, I choose to maintain my score (Accept).

---

> > > ### Author Response · Authors · 2024-08-13
> > > **Response to Reviewer EWje**
> > >
> > > We sincerely appreciate your valuable feedback and constructive comments. We are truly thankful for your recognition of our work.

---

### Author Rebuttal · Authors · 2024-08-05

We sincerely thank all the reviewers, ACs, SACs, and PCs for their effort and attention. We have uploaded a rebuttal file in PDF format to illustrate the tables and figures.

Table 1 presents the experimental results of the ensemble for four tasks, i.e., low-light image enhancement (LLIE), dehazing, deblurring with two base models, and deblurring with four base models. Table 2 measures the time cost of each step. Figure 1 and 2 show the visual comparisons of ensemble for LLIE and dehazing, respectively.

Due to space limitations, we list the newly cited works, including LOLv1 [1], OTS [2], RetinexFormer [3], RQ-LLIE [4], CIDNet [5], MixDehazeNet [6], DEA-Net [7], C2PNet [8], and NAFNet [9].


[1] Chen Wei, Wenjing Wang, Wenhan Yang, and Jiaying Liu. Deep retinex decomposition for low-light enhancement. arXiv preprint arXiv:1808.04560, 2018.

[2] Boyi Li, Wenqi Ren, Dengpan Fu, Dacheng Tao, Dan Feng, Wenjun Zeng, and Zhangyang Wang. Benchmarking singleimage dehazing and beyond. IEEE TIP, 28(1):492–505, 2018.

[3] Yuanhao Cai, Hao Bian, Jing Lin, Haoqian Wang, Radu Timofte, and Yulun Zhang. Retinexformer: One-stage retinex-based transformer for low-light image enhancement. In ICCV, 2023.

[4] Yunlong Liu, Tao Huang, Weisheng Dong, Fangfang Wu, Xin Li, and Guangming Shi. Low-light image enhancement with multi-stage residue quantization and brightness-aware attention. In ICCV, 2023.

[5] Yixu Feng, Cheng Zhang, Pei Wang, Peng Wu, Qingsen Yan, and Yanning Zhang. You only need one color space: An efficient network for low-light image enhancement. arXiv preprint arXiv:2402.05809, 2024.

[6] LiPing Lu, Qian Xiong, DuanFeng Chu, and BingRong Xu. Mixdehazenet: Mix structure block for image dehazing network. arXiv preprint arXiv:2305.17654, 2023.

[7] Zixuan Chen, Zewei He, and Zhe-Ming Lu. Dea-net: Single image dehazing based on detail-enhanced convolution and content-guided attention. IEEE TIP, 2024.

[8] Yu Zheng, Jiahui Zhan, Shengfeng He, Junyu Dong, and Yong Du. Curricular contrastive regularization for physics-aware single image dehazing. In CVPR, 2023.

[9] Liangyu Chen, Xiaojie Chu, Xiangyu Zhang, and Jian Sun. Simple baselines for image restoration. In ECCV, 2022.

---

### Decision · Program_Chairs · 2024-09-25

**Decision:**

Accept (poster)

**Comment:**

The reviewers agreed that the paper is well written and presents a novel method for ensemble-based prediction. The method has theoretical support, and the evaluation is extensive and demonstrates a consistent improvement.
There were some concerns about the resulting improvement being only marginal, and the limited practicality of the approach given the complexity of running many different base models, however the reviewers in general were in favour of accepting the paper.